# Three-dimensional atomic mapping of ligands on palladium nanoparticles by atom probe tomography

Kyuseon Jang [1,6], Se-Ho Kim [1,2,5,6], Hosun Jun [1], Chanwon Jung [1], Jiwon Yu [3], Sangheon Lee [3,4 ✉] & Pyuck-Pa Choi [1 ✉]

Capping ligands are crucial to synthesizing colloidal nanoparticles with functional properties. However, the synergistic effect between different ligands and their distribution on crystallographic surfaces of nanoparticles during colloidal synthesis is still unclear despite powerful spectroscopic techniques, due to a lack of direct imaging techniques. In this study, atom probe tomography is adopted to investigate the three-dimensional atomic-scale distribution of two of the most common types of these ligands, cetrimonium ($C_{19}H_{42}N$) and halide (Br and Cl) ions, on Pd nanoparticles. The results, validated using density functional theory, demonstrate that the Br anions adsorbed on the nanoparticle surfaces promote the adsorption of the cetrimonium cations through electrostatic interactions, stabilizing the Pd {111} facets. In contrast, the Cl anions are not strongly adsorbed onto the Pd surfaces. The high density of adsorbed cetrimonium cations for Br anion additions results in the formation of multiple-twinned nanoparticles with superior oxidation resistance.

[1] Department of Materials Science and Engineering, Korea Advanced Institute of Science and Technology (KAIST), Daejeon, Republic of Korea. [2] Department of Microstructure Physics and Alloy Design, Max-Planck-Institut für Eisenforschung GmbH, Düsseldorf, Germany. [3] Department of Chemical Engineering and Materials Science, Ewha Womans University, Seoul, Republic of Korea. [4] Graduate Program in System Health Science and Engineering, Ewha Womans University, Seoul, Republic of Korea. [5] Present address: Department of Microstructure Physics and Alloy Design, Max-Planck-Institut für Eisenforschung GmbH, Düsseldorf, Germany. [6] These authors contributed equally: Kyuseon Jang, Se-Ho Kim. ✉email: sang@ewha.ac.kr; p.choi@kaist.ac.kr

More than three decades of intensive research have fundamentally changed the perception of nanomaterials. What started as a scientific curiosity has led to novel materials, which are now used in real-world applications and have become essential for our everyday life. The research on nanomaterials was largely driven by their intriguing physical and chemical properties, such as the outstanding luminescence of semiconductor nanocrystals[1], the excellent energy storage performance of two-dimensional nanosheets[2], and the remarkable catalytic activity and selectivity of noble metal nanocatalysts[3,4]. These properties are usually ascribed to the limited size and high surface-to-volume ratio of nanomaterials, compared with their bulk counterparts.

Among the several nanomaterials reported to date, colloidal nanoparticles (NPs) are particularly attractive since they can be synthesized in large quantities at reasonable costs and their size, shape, and properties can be carefully tailored through optimized growth recipes[5]. Colloidal NPs are nowadays deployed on an industrial scale in key technological fields such as catalysis[6], energy conversion[7], and optoelectronics[8].

Various methods have been developed for synthesizing colloidal NPs, including hydrothermal synthesis[9], polyol synthesis[10], microemulsion technique[11], and sol–gel process[12]. Although each of these methods is unique, they share a common feature, i.e., they all rely on capping ligands. Capping ligands are additives adsorbed on specific crystallographic surfaces of the NPs; they can prevent NP agglomeration and control the NP size, shape, and functionality[13–16]. Therefore, capping ligands are paramount to tune the properties of colloidal NPs[17,18]. The most used capping ligands are thiols, block copolymers, cetrimonium, and halide ions[19]; the latter two are particularly advantageous as they can be applied in various NP systems[20–22].

Although capping ligands are indispensable for synthesizing colloidal NPs, little is known about their adsorption behavior on different crystallographic facets, especially at the atomic scale. There are still some important unanswered questions such as: What are the amounts of ligand molecules adsorbed on the NP surfaces? What is the interplay between different ligands added together during a growth process? How do ligands stabilize the NP surfaces thermodynamically and kinetically? How do ligands influence the inherent vulnerability of the NP surfaces against chemical attacks?

While there have been advances in analyzing capping ligands via nuclear magnetic resonance spectroscopy[23], scanning tunneling microscopy[24,25], transmission electron microscopy (TEM)[26,27], Fourier transform infrared (FT-IR) spectroscopy[28], and computational simulations[29], the direct mapping of the three-dimensional (3D) distribution, as well as the quantification of the ligands on the NPs remains a great challenge. The lack of experimental data is attributed to limited spatial resolution and/or detection sensitivity of many of the analytical techniques used[30].

Atom probe tomography (APT) is an advanced technique that can overcome such limations[31]. Its principle is based on the field evaporation of atoms from a needle-shaped specimen cryogenically cooled under an ultrahigh vacuum and subjected to an intense electric field (~10–50 V/nm). The specimen atoms undergo ionization during the field evaporation process; the resulting ions are accelerated by electrostatic forces toward a position-sensitive detector that measures their time-of-flight and impact coordinates, which are used for calculating a mass spectrum and reconstructing a 3D atom map, respectively.

The unique combination of near-atomic resolution and ppm-level detection sensitivity, irrespective of the elemental mass, makes APT an ideal tool for the characterization of nanomaterials. Therefore, we adopted this state-of-the-art technique to investigate the 3D distribution of cetrimonium ligands on

multiple-twinned NPs (MTNPs) of Pd, which are promising nanocatalysts for technologically important chemical reactions, such as oxygen reduction and formic acid oxidation, but are prone to oxidative eteching[32,33].

In this work, we synthesize highly stable Pd MTNPs via the simple reduction of a precursor in an aqueous solution by adding both cetrimonium cations ($C_{19}H_{42}N^+$, denoted as $CTA^+$ in the following) and Br anions. However, we also observe that replacing the Br anions with Cl anions reduces the yield of Pd MTNPs and lead to substantial oxidative etching of their surfaces; to understand this peculiar behavior, we analyze the distribution of $CTA^+$ on the Pd NPs via APT and directly attribute the formation of multiple-twinned structures and the oxidation resistance of the NP specimens to their surface coverage by $CTA^+$. To validate and support the experimental data, we calculate the binding energy of the ligands on various Pd NP facets by using the ab initio density functional theory (DFT). We demonstrate that the complex interplay between Pd NP surfaces, ligand ions, and solvent determines the shape and oxidative etching resistance of the final NPs.

## Results

**Structural characterization.** Figure 1 shows the TEM images of the Pd NPs synthesized by adding Br⁻ or Cl⁻ additions (hereafter, denoted as $Pd_{(Br)}$ and $Pd_{(Cl)}$ NPs, respectively). The $Pd_{(Br)}$ NPs mainly exhibited either icosahedral or decahedral (Fig. 1a and Supplementary Fig. 1) multiple-twinned structures, as confirmed via high-resolution TEM (HRTEM) and fast Fourier transform (FFT) imaging (see Fig. 1b for an icosahedral MTNP); all the facets showed a lattice spacing of 0.222 nm, corresponding to the interplanar spacing of the {111} planes (Fig. 1b)[34].

The MTNPs accounted for about 80% of the $Pd_{(Br)}$ NP batch, while the rest included minor products such as single-twinned right bipyramids, cubes, nanorods, and tetrahedra. The icosahedral NPs exhibited a smaller average size ($19 \pm 1$ nm) than the decahedral ones ($24 \pm 1$ nm); this result agrees with previous computational studies predicting that decahedra tend to form larger particles than icosahedra due to the more abundant twin boundaries of the latter and the correspondingly higher lattice strain energy[35].

Although twin boundaries with numerous active sites can enhance the catalytic properties of MTNPs, they seem also prone to oxidative dissolution[36,37]. Thus, the synthesis and the storage of MTNPs in oxidative environments, such as air or aqueous solutions, are challenging[38]. However, in this work, we made some unexpected observations for the $Pd_{(Br)}$ MTNPs. First, most of them did not undergo any remarkable oxidative dissolution although they were synthesized in an oxidative environment, i.e., in an aqueous solution at near-boiling temperature (90 °C) for 48 h in air. Moreover, they were largely preserved even after long-term storage in an oxidative environment, as discussed below. Besides, most of the $Pd_{(Br)}$ NPs exhibited only {111} facets, although Br anions were previously reported to promote the formation of {100} facets in Pd[39,40].

Unlike the $Pd_{(Br)}$ NPs, the $Pd_{(Cl)}$ NPs were mainly round-cornered cuboctahedra and tetrahedra (Fig. 1d, e), many of which showed strongly distorted shapes. Only about 30% of the $Pd_{(Cl)}$ NPs were MTNPs, whose corners and edges were rounder than those of the $Pd_{(Br)}$ MTNPs. Figure 1e shows an HRTEM and an FFT image of a $Pd_{(Cl)}$ NP having distorted shape, with no multiple-twinned structures; besides {111} facets, this distorted NP exhibited also {220} and {200} facets[41]. These observations indicate that the $Pd_{(Cl)}$ MTNPs underwent oxidative etching during their synthesis, while the $Pd_{(Br)}$ MTNPs were little affected by such a phenomenon.

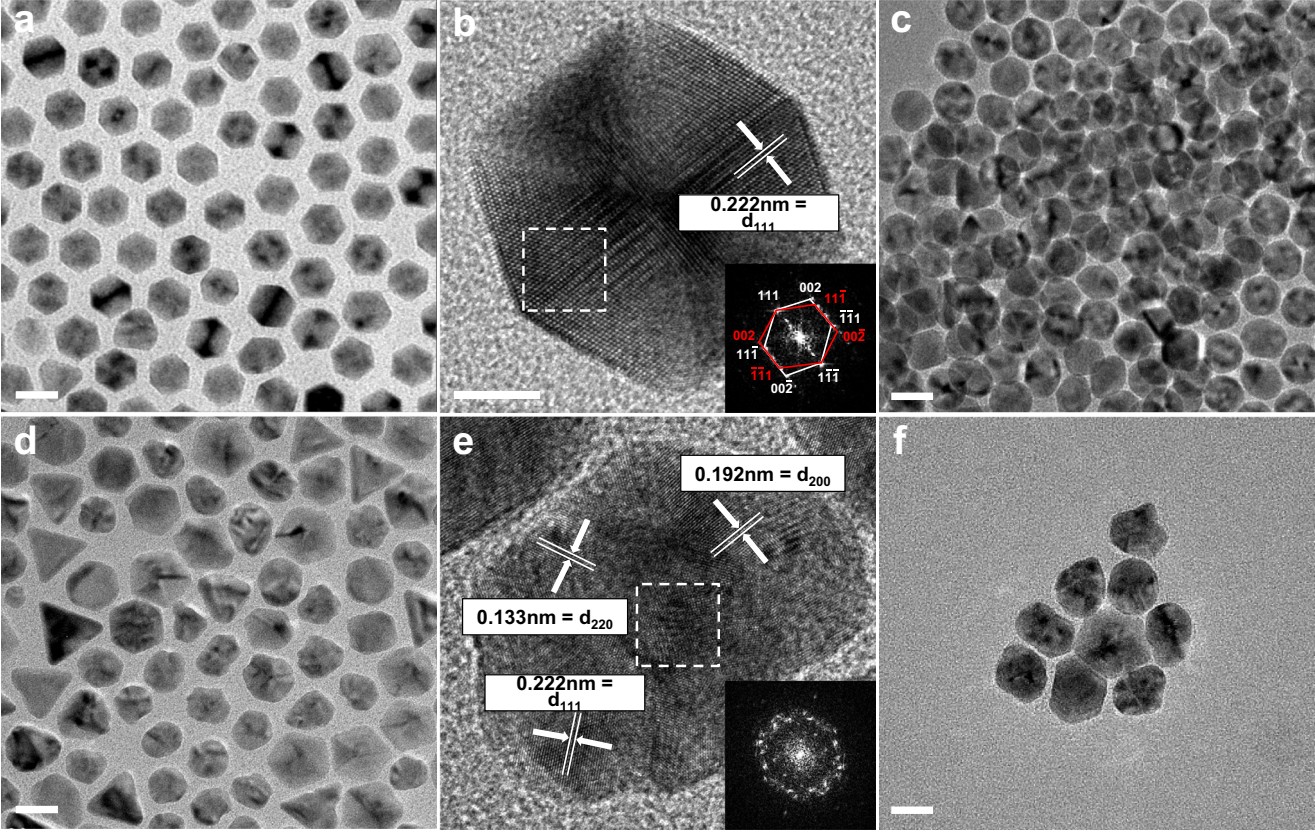

**Fig. 1 Transmission electron microscopy (TEM) images of the synthesized Pd nanoparticles (NPs). a–c** Multiple-twinned NPs produced by adding Br anions: **a**, **b** As-synthesized and **c** washed NPs after exposure to air for 10 days. **d–f** NPs produced by adding Cl anions: **d**, **e** As-synthesized and **f** washed NPs after exposure to air for 10 days. The insets in **b** and **e** show the fast Fourier transform patterns, which were recorded from the areas marked by the dashed squares. (Scale bars: 20 nm in **a**, **c**, **d**, **f** and 5 nm in **b**, **e**).

To further compare the resistance of the two NP batches against oxidative etching, we washed the $Pd_{(Br)}$ and $Pd_{(Cl)}$ colloidal specimens to remove excess $CTA^+$ and stored them at room temperature in air for 10 days. Since the $Pd_{(Cl)}$ NPs were exposed to a higher concentration of $Cl^-$ than the $Pd_{(Br)}$ ones during their synthesis, we added KCl to the $Pd_{(Br)}$ specimen before storage to match the amount of Cl anions acting as an etchant[40]. As shown in Fig. 1c, the $Pd_{(Br)}$ MTNPs retained most of their twin boundaries, showing slightly rounded vertices due to oxidative etching. Compared to previous studies, where the MTNPs were completely dissolved during synthesis in air[42], these $Pd_{(Br)}$ MTNPs exhibited remarkable stability against oxidative etching even for the tested long-term exposure to an oxidative environment (see Supplementary Fig. 2a). In contrast, less than 15% of the $Pd_{(Cl)}$ NPs were preserved (Fig. 1f and Supplementary Fig. 2b).

Furthermore, to consider the difference in oxygen solubility due to different salt concentrations in the $Pd_{(Br)}$ and $Pd_{(Cl)}$ specimens, we performed another set of control experiments. We added identical molar concentrations of KCl to the washed $Pd_{(Br)}$ and KBr to the washed $Pd_{(Cl)}$ NPs to match the salt concentrations and examined the specimens with TEM after 10 days of exposure to air. We made very similar observations as in the previous experiments, namely the $Pd_{(Br)}$ NPs were well preserved, whereas the $Pd_{(Cl)}$ NPs were largely dissolved (see Supplementary Fig. 3). Thus, the difference in oxidation resistance of the $Pd_{(Br)}$ and $Pd_{(Cl)}$ NPs could be indeed ascribed to the difference in concentrations of ligands on the Pd surface.

To determine whether these results were related to the varying amounts of adsorbed $CTA^+$, depending on the halide ion ligand ($Br^-$ or $Cl^-$) used, we also performed an FT-IR spectroscopy analysis. A reference $CTA^+/Cl^-$ specimen and the as-synthesized $Pd_{(Br)}$ and $Pd_{(Cl)}$ NPs, before and after a washing process for removing excess $CTA^+$ in the solutions, were analyzed (Supplementary Fig. 4). The FT-IR spectra of both as-synthesized specimens exhibited several peaks related to the $CTA^+$; the peaks at 3016, 1463, and 1375 $cm^{-1}$ were due to the asymmetric $-CH_3$ stretching bands, while those at 2917 and 2850 $cm^{-1}$ corresponded to the asymmetric $-CH_2-$ stretching bands. Furthermore, the peak at 719 $cm^{-1}$ was assigned to the $-CH_2-$ rock vibration[43] and those at 962 and 912 $cm^{-1}$ were attributed to the $-CN-$ vibration of the cationic head of the $CTA^+$ [43]. After washing the samples by centrifugation, the intensity of all the $CTA^+$-related peaks decreased by about 90%. However, the FT-IR data revealed the presence of residual $CTA^+$ on both the $Pd_{(Br)}$ and $Pd_{(Cl)}$ NPs even after washing, where the amount of $CTA^+$ was higher in the former ones.

**3D distribution and concentration of $CTA^+$ ligands on the Pd NPs.** We conducted APT measurements on washed $Pd_{(Br)}$ and $Pd_{(Cl)}$ NPs to reveal the 3D distribution and concentration of $CTA^+$ on their surfaces. For this analysis, we embedded the NPs in an electrodeposited Ni film via the method described in refs. 44,45; then, needle-shaped specimens were obtained from this composite Pd NP–Ni film through focused ion beam (FIB) milling[46].

Supplementary Figs. 5 and 6 show the mass spectra derived from the APT measurements of the as-prepared $Pd_{(Br)}$ and $Pd_{(Cl)}$ specimens, respectively. In both cases, the major isotope peaks were assigned to the Pd NPs and Ni in single- and double-charged states. $NiH^+$ and $NiO^+$ peaks were detected as well, possibly due to the presence of residual $H_2$ in the APT analysis

chamber and slight oxidation of the Ni film. We detected also $C^+$, $N^+$, $C_2^+$, $C_3^+$, and $C_4^+$ at 12, 14, 24, 36, and 48 Da, respectively. Moreover, other peaks were observed at 42, 43, and 44 Da, assigned to $C_2H_xN^+$ [47,48]. The spectrum of the $Pd_{(Cl)}$ specimen showed additional peaks between 80 and 100 Da, which could be assigned to complexes of Ni, C, and O, such as $NiC_x$ and $NiO_x$. To determine the amount of C or N impurities introduced from the electrodeposition process, we analyzed also bare electro-deposited Ni films; the measured average concentrations of C and N were only $0.010 \pm 0.005$ at.% and $0.012 \pm 0.003$ at.% (about 100 times lower than in the Pd-containing specimens), respectively (Supplementary Fig. 7).

To further clarify the origin of the C and N atoms detected in the $Pd_{(Br)}$ and $Pd_{(Cl)}$ specimens, we prepared Pd NPs without any $CTA^+$ ligands. APT specimens were prepared from these NPs, using the same procedure as for the $Pd_{(Br)}$ and $Pd_{(Cl)}$ specimens. As can be seen in the acquired APT data (see Supplementary Fig. 8) the detected C concentration was diminishingly low (~35 ppm) as compared to the $Pd_{(Br)}$ and $Pd_{(Cl)}$ specimens. Moreover, no significant C segregation was detected on the Pd surface. Thus, we conclude that the C segregation zones detected for the $Pd_{(Br)}$ and $Pd_{(Cl)}$ specimens can be indeed ascribed to the $CTA^+$ ions.

The detected halide ion concentrations were very low in both $Pd_{(Br)}$ and $Pd_{(Cl)}$ specimens, namely <0.008 at.% Br in $Pd_{(Br)}$, <0.024 at.% Cl in $Pd_{(Br)}$, and <0.069 at.% Cl in $Pd_{(Cl)}$ (See Supplementary Fig. 9). This difference in the concentration levels can be ascribed to the fact that Cl was present in three reagents ($K_2PdCl_4$, as the precursor for Pd NPs, cetrimonium chloride, and KCl ligands; see the Methods section) while Br only in one reagent (KBr).

Additionally, the level of implanted $Ga^+$ ions was estimated by analyzing the bulk mass spectra of cuboidal regions of interest (ROIs) ($40 \times 40 \times 40$ nm³ in size) containing the Pd NPs and surrounding ligands. Taking into account the peak overlap between the $Ga^+$ and $Ni_2O^+$ ions at 69 Da (see Supplementary Fig. 10), the maximum Ga concentrations around the NPs in the $Pd_{(Br)}$ and $Pd_{(Cl)}$ specimens were estimated to be 0.02 and 0.07 at.%, respectively. Figure 2a illustrates an APT reconstruction containing two $Pd_{(Br)}$ NPs embedded in Ni, along with an iso-concentration surface of 1.5 at.% C. At the reconstructed Pd/Ni interface, the C concentration was about 3 at.%; the C atoms were detected on the surfaces of both the top and bottom Pd NPs. The 10 nm thin slice viewed along the z-axis shown in Fig. 2b reveals a projection of the top Pd NP. The bottom Pd NP, which was partly detected, exhibited a corner, as shown in the slice viewed along the x-axis

displayed in Fig. 2c. Both Pd NPs showed segregation of C and N atoms at their surfaces. Supplementary Fig. 11 illustrates another APT dataset where a part of a Pd NP, fully covered by C, was detected.

One $CTA^+$ ($C_{19}H_{42}N^+$) consists of one N atom bonded to three methyl- and one hexadecyl-carbon group, resulting in a C:N ratio of 19:1. The average bulk compositions of three different regions containing C-complexes (marked by the C iso-concentration surface) showed a C:N ratio of 18.9:1 (see Supplementary Fig. 12 and Supplementary Table 1), confirming that the C and N atoms indeed originated from the $CTA^+$ ions. Thus, these APT results indicate the segregation of $CTA^+$ on the surface of the $Pd_{(Br)}$ NPs.

Figure 3a displays a reconstructed 3D atom map of three $Pd_{(Cl)}$ NPs in Ni, along with an iso-concentration surface of 3 at.% C. Unlike the $Pd_{(Br)}$ NPs, the C atoms were detected not only at the Pd surface but also deep within the Ni matrix, i.e., about 10 nm away from the $Pd_{(Cl)}$ NPs (Fig. 3b, c). Similar results were observed for the Cl atoms.

To quantify the amount of $CTA^+$ adsorbed on the surfaces of the $Pd_{(Br)}$ and $Pd_{(Cl)}$ NPs, we calculated the Gibbsian interfacial excess of the C atoms at the Pd/Ni interface ($\Gamma_C$). For a cylindrical ROI ($\varnothing 5 \times 30$ nm³) aligned perpendicularly to a surface of a reconstructed Pd NP, the corresponding $\Gamma_C$ can be expressed as[49]

$$\Gamma_C = \frac{1}{A\eta} N_C^{excess} = \frac{1}{A\eta} \left( N_C^{total} - N_C^{Pd} - N_C^{Ni} \right), \qquad (1)$$

where A and $\eta$ are the cross-sectional area of the cylindrical ROI (19.6 nm²) and the detection efficiency (37%) of the APT instrument[50], respectively, $N_C^{excess}$ is the excess number of C atoms at the Pd/Ni interface, $N_C^{total}$ is the number of total C atoms in the ROI, and $N_C^{Pd}$ and $N_C^{Ni}$ are the numbers of C atoms in bulk Ni and Pd, respectively. Since $N_C^{Pd}$ and $N_C^{Ni}$ were below the background noise level of the mass spectra of the $Pd_{(Br)}$ and $Pd_{(Cl)}$ NPs, we considered them to be zero for simplicity. The concentration of C impurities within the bare electrodeposited Ni films was just above the detection limit ($0.010 \pm 0.005$ at.%), and therefore, we assumed that all the C atoms originated from the $CTA^+$. From the $\Gamma_C$ value, we could derive the surface excess of $CTA^+$ on the Pd NPs ($\Gamma_{cetrimonium}$, in molecular ions per nm²) as follows

$$\Gamma_{cetrimonium} = \Gamma_C \times \frac{1 \text{ cetrimonium molecule}}{19 \text{ carbon atoms}}, \qquad (2)$$

where the second factor indicates that one $CTA^+$ contains 19 C atoms. For both $Pd_{(Br)}$ and $Pd_{(Cl)}$ specimens, eight cylindrical ROIs

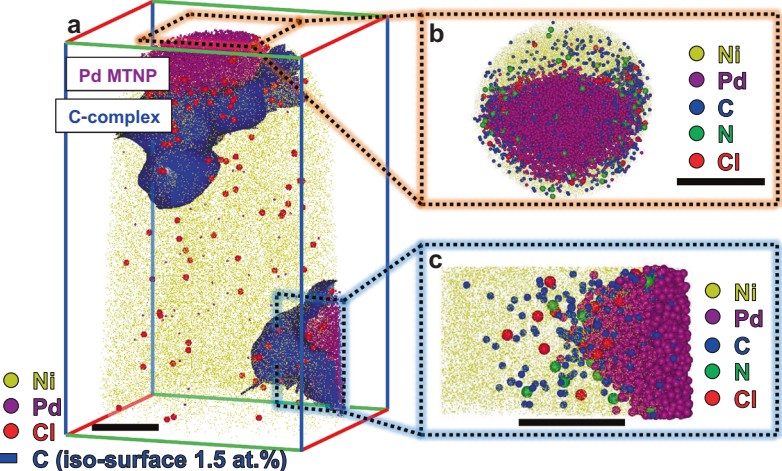

**Fig. 2 Atom probe tomography (APT) reconstruction of Pd multiple-twinned NPs (MTNPs), synthesized by adding Br anions, embedded in Ni. a** Three-dimensional atom map of Pd atoms and iso-concentration (1.5 at.%) surfaces of C. **b, c** Slices viewed along the Pd MTNPs. (Scale bars: 10 nm).

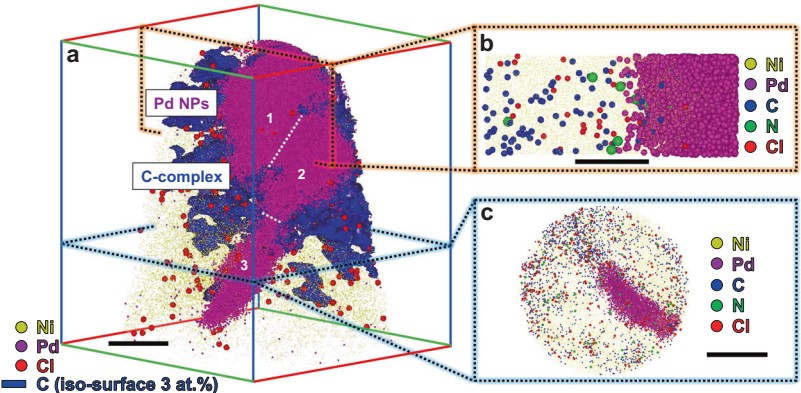

**Fig. 3 Atom probe tomography (APT) reconstruction of Pd NPs, synthesized by adding Cl anions, embedded in Ni. a** Three-dimensional atom map of Pd atoms and iso-concentration (3 at.%) surfaces of C; the white dotted lines indicate the boundaries of each NP. **b, c** Slices viewed along with the Pd NPs. (Scale bars: 10 nm in **a**, 5 nm in **b**, 20 nm in **c**).

were placed at different locations across the interface between an NP and the matrix. The corresponding C excess values were determined from a cumulative plot of the C atoms against the total number of atoms within the ROI according to the method proposed by Krakauer et al.[49]. Further details on this method are given in the supporting information, where the results are listed in Supplementary Table 2 and shown as a scatter plot in Supplementary Fig. 13. The determined average values of the cetrimonium surface density were $1.9 \pm 0.2$ and $0.7 \pm 0.3$ CTA$^+$/nm$^2$ for the Pd$_{(Br)}$ and the Pd$_{(Cl)}$ NPs, respectively. While the data points are more scattered in the case of Pd$_{(Cl)}$, the difference in the CTA$^+$ surface density between Pd$_{(Br)}$ and Pd$_{(Cl)}$ is clearly beyond the error range (see Supplementary Table 2 and Supplementary Fig. 13). The acquired APT results could be qualitatively confirmed by thermogravimetric analyses (TGA) and X-ray photoelectron spectroscopy (XPS) on washed Pd$_{(Br)}$ and Pd$_{(Cl)}$ NPs (see Supplementary Figs. 14, 15, and Supplementary Table 3). Furthermore, these results were also supported by CO-DRIFT measurements performed on washed Pd$_{(Br)}$ and Pd$_{(Cl)}$ NPs, where more CO molecules were adsorbed on the Pd$_{(Cl)}$ NPs than on the Pd$_{(Br)}$ NPs due to a lower amount of adsorbed CTA$^+$ (Supplementary Fig. 16 and Supplementary Table 4).

The hydrophilic cationic head of a CTA$^+$ molecular ion consists of one N atom and three methyl (CH$_3$) groups and spans over an area of 0.64 nm$^2$ [51], indicating that 1.6 CTA$^+$ are required to completely cover 1 nm$^2$ of a Pd {111} surface on average. Thus, our findings indicate that the surface of a Pd$_{(Br)}$ NP can be completely covered with CTA$^+$.

We could determine the $\Gamma_{cetrimonium}$ value for the Pd$_{(Cl)}$ specimen in an identical manner for the regions where the C atoms were concentrated. The maximum value was 1.2 CTA$^+$ per nm$^2$, indicating that the surface of a Pd$_{(Cl)}$ NP cannot be completely covered with CTA$^+$, in contrast with the Pd$_{(Br)}$ NPs. These results imply that the type of halide ions used in the NP synthesis can influence the adsorption behavior of the surfactant molecules on the NPs. Moreover, the difference in the number density of CTA$^+$ on the particle surfaces explains why the Pd$_{(Br)}$ MTNPs showed remarkable stability whereas the Pd$_{(Cl)}$ NPs were prone to oxidative etching; the CTA$^+$ completely covered the surfaces of the Pd$_{(Br)}$ MTNPs, protecting them from being etched by oxidative reactants such as Cl$^-$, OH$^-$, and O$_2$ during their synthesis and storage.

**Theoretical discussion of the binding behavior of ligands on Pd NPs.** To validate the APT results and clarify the adsorption behavior of ligands on the studied Pd NPs, we performed DFT calculations. First, we calculated the surface energy for the low-

indexed facets of pure Pd, i.e., the {100}, {110}, and {111} surfaces (Fig. 4a), observing an increase in the as-obtained values in the order of {111} (0.085 eV/Å$^2$), {100} (0.094 eV/Å$^2$), and {110} (0.101 eV/Å$^2$), which is in good agreement with previous theoretical and experimental results[52]. Thus, if the shape of a Pd NP is determined only by its surface energy, the Wulff construction predicts that a Pd NP without any adsorbed ligands will form {111} and {100} facets[40,53].

Next, we calculated the binding energy of the Br and Cl anions on each Pd facet in a vacuum; regardless of the facet, Cl$^-$ exhibited stronger interaction with the Pd surfaces than Br$^-$ (Fig. 4b). A detailed analysis of the projected density of states (PDOS) showed that the $p$-orbitals of the halide ions were overlapped with the $d$-orbitals of the Pd surfaces, and this overlap was stronger for the Cl anions than the Br ones, forming stronger covalent bonds (Supplementary Fig. 17).

However, since the actual synthesis of Pd$_{(Br)}$ and Pd$_{(Cl)}$ NPs was performed in aqueous solutions, the binding behavior of the halide ions on Pd facets in an aqueous medium had to be considered. Hence, we derived the binding energy in solutions from the values for vacuum according to the Born–Haber cycle[54] (Supplementary Fig. 18). As shown in Fig. 4c, the Br anions showed stronger interactions with the Pd {100}, {110}, and {111} facets than the Cl ones. This result was mainly attributed to the lower solvation energy and electron affinity of Br$^-$ compared to Cl$^-$ (Fig. 4d, e)[55], whose lower solvation energy is due to the larger ionic radii of Br$^-$ (Cl$^-$: 1.81 Å; Br$^-$: 1.96 Å)[56].

Furthermore, the binding energy between Br$^-$ and Pd NPs was anisotropic. Figure 4e also shows that the Br anions bound more strongly to the {100} and {111} facets (–1.821 and –1.805 eV, respectively) than to the {110} ones (–1.579 eV) of Pd NPs in solutions, although the latter ones exhibited the highest $d$-orbital center and, hence, were expected to be more reactive than the others (Supplementary Fig. 19). This contradictory result was attributed to the higher work function ($\Phi_{Pd}$) values of the {100} and {111} (5.180 and 5.405 eV, respectively) than {110} surfaces (4.935 eV) (Fig. 4f), which led to higher stability compared to the {110} facets upon adsorbing Br$^-$. However, since the Pd {111} and {100} facets have similar surface energy and Br$^-$ binding energy, the above considerations cannot fully explain the exclusive formation of {111} facets in Pd$_{(Br)}$ MTNPs.

To further clarify the adsorption behavior of Br$^-$ on Pd, we calculated the binding energies of Br anions on different adsorption sites (face-centered cubic (FCC), hexagonal closed-packed (HCP), bridge, top, and hollow) of the Pd {111} and {100} surfaces (Fig. 4g). For the {111} facets, we identified four representative adsorption sites, i.e., FCC, HCP, bridge, and top

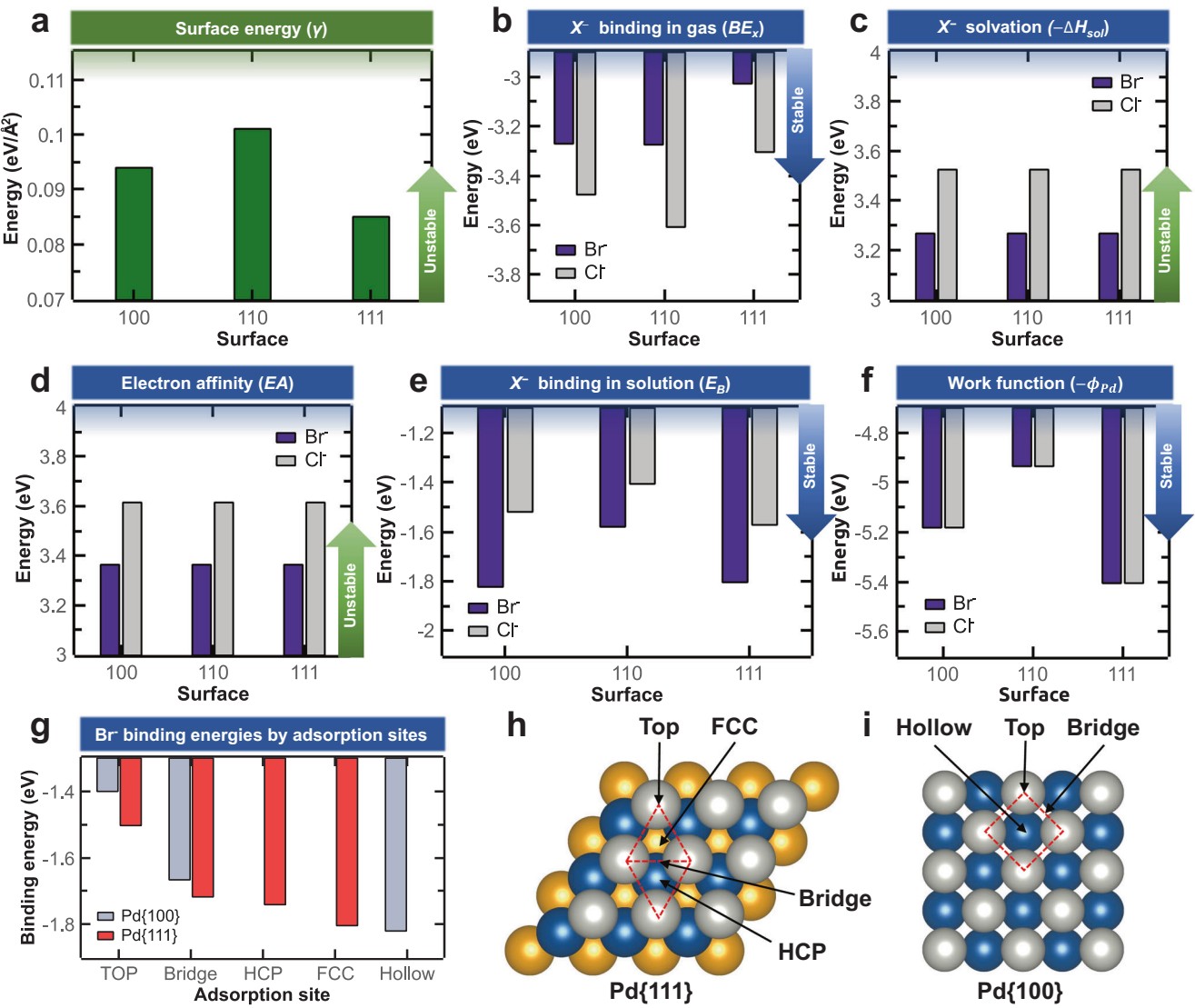

**Fig. 4 Density functional theory calculations. a** Surface energy values for Pd {100}, {110}, and {111} facets. **b** Binding energies of halide anions on Pd in a vacuum. **c** Solvation energies of halide anions. **d** Electron affinities of halide anions. **e** Binding energies of halide anions on Pd in solutions. **f** Work functions of Pd surfaces. **g** Binding energies of Br⁻ for the different adsorption sites of Pd surfaces; **h, i** Corresponding adsorption sites (face-centered cubic (FCC), hexagonal close-packed (HCP), bridge, and top) for Pd {111} and {100} facets from a top view.

(Fig. 4h)[57,58]. The binding strength increased in the following order: top (−1.504 eV) < bridge (−1.717 eV) < HCP (−1.741 eV) < FCC (−1.805 eV). For the {100} facets, we found three representative adsorption sites, that is, hollow, bridge, and top (Fig. 4i)[57,58], and the binding strength increased as follows: top (−1.400 eV) < bridge (−1.667 eV) < hollow (−1.821 eV). The PDOS analysis revealed that these site-specific variations of binding strength for both facets were related to the covalent degree of the bonding between Pd surface and Br⁻ through the overlap of the $p$-orbitals of the latter and the $d$-orbitals (especially the $d_{xz}$ and $d_{yz}$ orbitals) of the former (Supplementary Fig. 20). These results indicate that the Pd {111} facets comprise more energetically favorable adsorption sites than the {100} ones and, thus, can accommodate a larger number of Br anions.

## Discussion

The different adsorption behavior of CTA⁺ on the Pd$_{(Br)}$ and Pd$_{(Cl)}$ NPs can be ascribed to the different interactions of Br⁻ and Cl⁻ with the Pd surfaces. The halide anions locally chemisorbed

to a Pd surface usually form a negatively charged layer, attracting positively charged CTA⁺ and forming bonds with them via electrostatic interactions[52,59,60]. The DFT results showed (Fig. 4e) that the Br anions can be adsorbed on the Pd surfaces with a higher density than the Cl ones in solution; we also found that Br⁻ can promote more CTA⁺ adsorption than Cl⁻, consistent with previous molecular dynamics simulations for Au surfaces[61,62].

A comparative analysis of the Br⁻ binding energy for various adsorption sites on the Pd {111} and {100} facets (Fig. 4g) suggested that the {111} surfaces exhibit the highest number density of adsorption sites. Therefore, the Br⁻ layers formed on a {111} surface are expected to show a higher charge density than those formed on a {100} facet, allowing a higher density of bonds with CTA⁺. Since the cationic heads and the alkyl tails of CTA⁺ are hydrophilic and hydrophobic, respectively, the CTA⁺ may be adsorbed on the Pd surfaces via their cationic heads and form a double layer with the cationic heads of the second layer facing outward (Fig. 5)[63]. The Br⁻ layer formed on the Pd$_{(Br)}$ NP

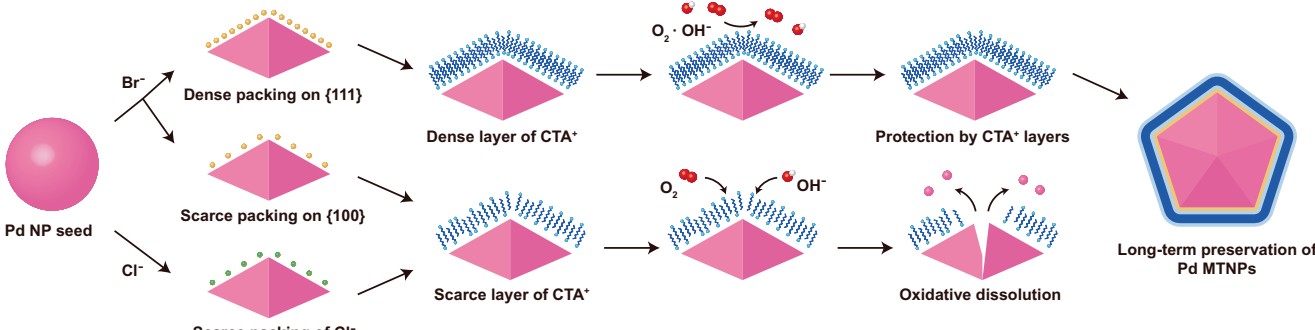

**Fig. 5 Influence of halide anions and CTA$^+$ on the shape and oxidation stability of Pd NPs and MTNPs.** The density of Br$^-$ adsorbed on the Pd {111} facets is high, enhancing adsorption of CTA$^+$ through electrostatic forces and thereby yielding Pd MTNPs of high oxidation resistance. In contrast, the density of adsorbed Cl$^-$ is comparatively low, resulting in low adsorption of CTA$^+$ and oxidative etching of the Pd surface.

surfaces not only exerts an attractive electrostatic force on CTA$^+$ but may also partly screen the electrostatic repulsion between the cationic heads of CTA$^+$, enhancing their surface excess[63,64]. Thus, the high density of adsorbed CTA$^+$ stabilizes the {111} facets, in particular for the Pd$_{(Br)}$ MTNPs (Fig. 5). Moreover, Br$^-$ and CTA$^+$ may reduce the growth rate of the MTNPs, suppressing their evolution into single-crystal NPs[40].

These mechanisms explain why the MTNPs, comprising {111} facets only, were the major product with Br$^-$ and CTA$^+$ for Pd$_{(Br)}$. We note that the joint addition of Br$^-$ and CTA$^+$ is the key to the synthesis of MTNPs. As in our case, a previous study demonstrated that Au–Pd NPs with {111} facets could be synthesized by using both Br$^-$ and CTA$^+$ in aqueous solutions[65], whereas the addition of Br$^-$ alone can promote the formation of the {100} facets of noble metal NPs[39,40]. These results indicate that the shape control of noble metal NPs by Br$^-$ addition cannot be ascribed only to the ligand ion species but to the overall chemical environment during the NP synthesis, including the interactions between different ligands, as well as between ligand and solvent molecules, besides the adsorption behavior of the ligand ions on the NP surfaces.

Moreover, a different ligand adsorption behavior explains why the Pd$_{(Br)}$ NPs exhibited substantially higher resistance to oxidative etching than the Pd$_{(Cl)}$ ones. Capping ligands can protect MTNPs against oxidative attack by covering their surfaces[38,66]; our findings support this mechanism (Fig. 5). The detected number density of CTA$^+$ on Pd$_{(Cl)}$ NPs was insufficient to form a monolayer, while the CTA$^+$ fully covered the surfaces of the Pd$_{(Br)}$ NPs. Thus, the Pd$_{(Cl)}$ NPs were prone to oxidative etching, whereas the Pd$_{(Br)}$ MTNPs showed high stability even over long-term (10 days) storage in air.

In this work, we elucidated the adsorption and 3D distribution of ligands on colloidal Pd NPs via APT measurements and ab initio DFT calculations. We clarified the binding nature of ligands and the interplay between halide ions (Br$^-$ and Cl$^-$) and CTA$^+$, which are among the most commonly used ligands in colloidal synthesis. We revealed that under the given synthesis condition, Br$^-$ ions are more strongly chemisorbed on the Pd surface than Cl$^-$ ions and enhance the adsorption of CTA$^+$ ions on Pd NP through electrostatic interaction. The surface excess of CTA$^+$ was higher than the value required to form a monolayer when synthesizing Pd NPs with Br$^-$ addition, stabilizing the Pd {111} facets and multiple-twinned structures and enhancing the resistance of the Pd MTNPs against oxidative etching.

Finally, the direct imaging of ligands, as demonstrated here, should be extended to other systems to provide a general understanding of the ligand adsorption on NPs. This is essential for the knowledge-based tailoring of the physical and electrochemical properties of NPs for specific target applications.

## Methods

**Chemicals**. Potassium tetrachloropalladate (II) (K$_2$PdCl$_4$, 98 %), potassium bromide (KBr, 99 %), potassium chloride (KCl, 99 %), cetrimonium chloride (CTAC, Mw = 320.00, 25 wt% in water), all purchased from Sigma Aldrich, were used for the synthesis of Pd NP. Nickel sulfate heptahydrate (NiSO$_4$·6H$_2$O, Junsei Chemical Co.) and boric acid (H$_3$BO$_3$, Sigma Aldrich), and nickel chloride hexahydrate (NiCl$_2$·6H$_2$O, Samchun Chemical Co.) were used for the electrodeposition process. Deionized (DI) water was used in all experiments.

**Synthesis of Pd NPs**. Pd$_{(Br)}$ NPs were synthesized by using KBr. About 13 mg of K$_2$PdCl$_4$ (II), 48 mg of KBr, 0.1 ml of CTAC were dissolved in 7 ml of distilled water. K$_2$PdCl$_4$ and KBr were used as a precursor and a shape-controlling agent, respectively. CTAC was added to serve both as a surfactant and a mild reducing agent[67]. The prepared precursor solution was placed in an oven and kept at 90 °C for 48 h in air. During the reaction the solution color changed from turbid orange to black, indicating that the Pd precursor was reduced to form zero-valent Pd NPs. Pd$_{(Cl)}$ NPs were synthesized by replacing KBr with KCl and dissolving 30 mg of KCl instead of KBr while maintaining the other synthesis conditions.

**Embedding Pd NPs in a Ni matrix for APT sample preparation**. The Pd NPs were collected using a centrifuge (7041xg for 30 min) and re-dispersed in distilled water under ultrasonication for 30 min. This washing process was performed three times for removing excess amounts of CTAC from the solution and observing only the CTAC attached to the NPs.

As-washed Pd NPs were electrodeposited within a Ni matrix according to the procedure developed by Kim et al[44]. The process consisted of two steps, namely electrophoresis of NPs on a flat Cu substrate followed by electroplating of a Ni matrix.

A vertical cell with a Cu substrate placed at the bottom and a Pt electrode on top was used for both electrophoresis and electroplating using a potentiostat (WPG100e, WonATech). For electrophoresis, the Pd NP solution was poured into the cell and a constant current of 10 mA was applied for 100 s. Subsequently, the remaining NP solution was removed and replaced by a modified Watts solution for Ni plating[68]. Electrodeposition of Ni was carried out at a constant current of 100 mA for 200 s.

**Fourier transform infrared (FT-IR) spectroscopy analysis**. A Nicolet iS 50 FT-IR spectrometer (Thermo Scientific, USA) was used for the collection of spectra in the range from 400 to 3600 cm$^{-1}$ at a spectral resolution of 1.928 cm$^{-1}$. The data analysis was carried out using the OMNIC software (Version 9.2.106, Thermo Scientific, USA).

**Thermogravimetric analysis (TGA)**. A TG209 F1 Libra (NETZSCH) instrument was used for TGA measurements in the temperature range from 50 to 700 °C at a heating rate of 10 °C/min. All measurements were carried out in a nitrogen atmosphere.

**X-ray photoelectron spectroscopy (XPS) analysis**. A Kratos Axis-Supra instrument was used for XPS measurements using monochromatic Al Kα radiation (1486.7 eV). Photoelectrons were collected at a take-off angle of 90° relative to the sample surface. Data analyses were performed using the Thermo Scientific

Avantage software. The binding energies of the spectra were calibrated by setting the C–C binding energy of the C1s peak to 284.8 eV.

**CO-DRIFT measurements**. A Nicolet iS 50 FT-IR spectrometer (Thermo Scientific, USA) equipped with an in situ cell was used for CO-DRIFT measurements at room temperature. NPs dried at 50 °C for 12 h were pretreated at room temperature in a He environment for 1 h to record a background spectrum. Subsequently, a mixture of He gas and 1 vol% of CO was injected into the cell at a flow rate of 30 ml min$^{-1}$ until saturation. After purging with nitrogen (60 ml min$^{-1}$) to remove the physisorbed CO molecules, the DRIFT spectra were recorded at a resolution of 4 cm$^{-1}$.

**TEM and APT characterization**. As-synthesized NPs were characterized with respect to their size distribution and morphology using TEM (Tecnai G2 F30 S-Twin) operated at 300 kV in conventional mode. Average particle sizes were determined from 100 NPs randomly selected from TEM images. HAADF-STEM images were obtained on an FEI Talos F200X operated at 200 kV. TEM specimens were prepared by depositing a water-dispersed NP sample on a carbon-coated copper grid. APT specimens were prepared using FIB (Helios Nanolab 450, FEI) milling according to ref. [46,69]. In order to reduce the implantation of Ga$^+$ ions during FIB milling to a minimum level, we applied a final low-kV (5 kV) clean-up step for the sharpened APT specimens.

APT measurements were done using a CAMECA LEAP$^{TM}$ 4000X HR system in pulsed-laser mode at a detection rate of 0.3%, a base temperature of 65 K, a laser pulse energy of 50–60 pJ, and a pulse frequency of 125 kHz. Data reconstruction and analyses were performed using the commercial software, Imago visualization and analysis system (IVAS) 3.8.2 developed by CAMECA Instruments. All three-dimensional atom maps presented in this paper were reconstructed using the standard voltage reconstruction protocol[70].

**DFT calculations**. Spin-polarized DFT calculations were performed within the generalized gradient approximation (GGA-PW91), as implemented in the Vienna Ab-initio Simulation Package (VASP)[71]. The projector augmented wave (PAW) method with a plane wave basis set was employed to describe the interaction between ion cores and valence electrons[72]. An energy cutoff of 400 eV was applied for the expansion of the electronic eigenfunctions. Supplementary Fig. 21 shows the geometries of the surface models ({100}, {110}, and {111} planes of Pd) used for the DFT calculations. For constructing the supercell, each model surface was separated from its periodic images in the vertical direction by a vacuum space corresponding to ten atomic layers. For the Brillouin zone integration of Pd (100), (110), and (111) surfaces, we used a $(5 \times 5 \times 1)$ Monkhorst-Pack mesh of k points to determine the optimal geometries and total energies of systems[73]. For each surface model, all Pd atoms were fixed at corresponding bulk positions, while the adsorbate position was fully relaxed using the conjugate gradient method until residual forces on all the constituent atoms became smaller than $5 \times 10^{-2}$ eV/A. For the work function calculation, we increased the k point mesh to $(10 \times 10 \times 1)$.

The surface energy ($E_{Surf}$) of each facet was determined by the relation $E_{surf} = \frac{E_{slab} - N\epsilon}{2A}$, where $N$ is the number of total Pd atoms in a simulation box, $\epsilon$ is the cohesive energy per atom of the Pd bulk, $E_{slab}$ is the total energy of the surface structure with vacuum, and $A$ is the surface area.

According to the Born–Haber cycle approach (see Supplementary Fig. 18), the binding energy ($E_B$) of a halide ion ($X^-$) to a Pd surface is given as $E_B = BE_x - \Delta H_{sol} - \Phi_{Pd} + EA_X$, where $BE_x$ is the binding energy for the adsorption of halogen atoms to a negatively charged Pd surface with one extra electron, $\Delta H_{sol}$, $\Phi_{Pd}$, and $EA_x$ are the solvation energy of $X^-$ in liquid water, the work function of a corresponding Pd surface, and the electron affinity of X, respectively[52]. Here, $BE_x$ is given as $BE_x = E_{Pd-X} - E_s - E_X$, where $E_{Pd-X}$ is the energy of the negatively charged surface with the bound halogen atom, $E_s$ is the energy of the bare Pd surface without any adsorbate but with one extra electron, and $E_X$ is the energy of the neutral atom (X). We assumed that the interaction between water and Pd before and upon X adsorption does not change. $\Delta H_{sol}$ and $EA_X$ values were taken from the reports by Marcus et al. and NIST, respectively[74,75].

## Data availability

The data that support the findings of this study are available from the corresponding authors upon reasonable requests.

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

## Acknowledgements

K.J., S.-H.K., H.J., C.J., and P.-P.C. acknowledge the support of the National Research Foundation of Korea (NRF) (NRF-2019R1A2C1002165) and of the KAIST Venture Research Program for Graduate and Ph.D. students. J.Y. and S.L. acknowledge the support of the Basic Science Research Program through the National Research Foundation (NRF) of Korea funded by the Ministry of Education (NRF-2018R1D1A1B07051430). S.-H.K. acknowledges financial support from the ERC-CoG-SHINE-771602.

## Author contributions

K.J. and S.-H.K. designed all the experiment, carried out chemical synthesis, and characterization. H.J. and C.J. carried out TEM analysis and FIB processing. K.J., S.-H.K., H.J., and C.J. conducted APT analysis and data processing. J.Y. and S.L. performed DFT calculations. All authors discussed the results and contributed to the manuscript writing under the supervision of S.L. and P.-P.C.

## Competing interests

The authors declare no competing interests.
