## [Peer Review File · Nature Communications]

REVIEWER COMMENTS

Reviewer #1 (Remarks to the Author):

Authors attempt to characterize the halide ions and CTA+ ligands adsorbed on Pd nanoparticles using atom probe tomography and then use DFT calculations to validate the findings of lower CTA+ ligand coverage on Pd(Cl) versus Pd(Br). Then authors use both the APT and DFT results along with TEM results of air exposed nanoparticle stability to attribute the improved stability of Pd(Br) against oxidative etching on the higher coverage of CTA+ on Pd(Br) nanoparticles. While this work has some value as a first attempt to analyze organic ligands attached to nanoparticle by APT, I do have some serious concerns on the influence of sample preparation process on the entire APT experimental results presented here in this manuscript.

Specifically, once the CTA+ ligand coated Pd(Br) and Pd(Cl) nanoparticles are synthesized, they are then washed three times, centrifuged, then subjected to electrophoresis, then to electrodeposition to coat the Ni as a capping layer and then to FIB milling at room temperature to make the final APT needle. Shouldn't these steps influence the ligand distribution on nanoparticles rather substantially? These are rather intense mechanical, electrochemical, ion irradiation induced modification of the ligand layers before they are analyzed in APT. However, in this work authors show APT results from Pd(Br), Pd(Cl) and Pd nanoparticle free electrodeposited Ni coating. But I do feel authors have to show a result of clean Pd nanoparticles without Br or Cl or CTA+ but after that also went through the entire sample preparation process and show that there is no Carbonaceous species at the Ni-Pd nanoparticle interface to prove that the APT sample preparation is not what is leading to the carbon rich layer. Without this, directly attributing all the measured Carbon atoms to be coming only from CTA+ feels little bit like a leap of faith.

Also authors have to show all the details of the interfacial excess calculation from APT because that's one of the main point highlighted in this paper. Also the error value for the CTA+ excess in Pd(Cl) needs to be provided. I also have concern on how repeatable these carbon excess quantification are from particle to particles. So rather than just providing the average of carbon excess for 8 measurements, authors should provide a scatter plot with error bars to show is there a clear above error bar level difference in the measured carbon excess between Pd(Br) and Pd(Cl) to be conclusive in the results. Without that this result may not be repeatable.

Some more suggestions are

- The second statement in abstract "interaction between capping ligands and their interaction with the crystallographic surfaces of nanoparticles during colloidal synthesis remains a great mystery, due to the lack of direct imaging techniques." is a bit of an overstatement. There are great many spectroscopic techniques to analyze ligand interaction.
- Please provide the chemical formula of citrimonium in the abstract.

- The FTIR figure doesn't seem crucial for the manuscript and hence can be moved to supplementary information
- In supplementary figures 3 and 4 APT mass-to-charge spectra, the y axis labels should be log (counts)?
- I notice the spectra FWHM is deteriorated considerably in the case of Pd (Cl) versus Pd(Br) why is this? And how is this influencing the interpretations given in this work?
- Authors needs to show an overlaid comparison of Cl peaks and peaks at the locations where Br peaks should be present between the two samples. Given the high likelihood of substantial overlaps between organic peaks and Cl peaks as well as Br peaks there can be considerable uncertainty in assigning the peaks at 35 and 37 Da as well as peaks at 39.5 and 40.5Da peaks.
- About Figure 3 a and b authors talk about pentagonal projection and say this is a decahedral particle. Its difficult to see conclusively from the APT figures.
- In figure 3 it is recommended to show how Cl is distributed in the Pd(Br) samples also, since authors are showing evidence for clear peaks of Cl detected in the APT mass to charge spectra of Pd(Br) in supplementary figure 3(a) but it is missing in figure 3 of manuscript.
- In page 10 authors mention the C:N ratio was 1:18.4. To support this, it is recommended to show the cylinder line composition profile perpendicular to the Pd-Ni interface from where this was estimated in the supplementary data with error bars.
- Unclear why authors used 1.5 at% C isoconcentration surface in Pd(Br) in figure 3(a) and 3 at% isoconcentration surface in the case of Pd(Cl) in Figure 4(a)?
- The most highlighted finding in this paper is the interfacial excess of CTA+ of 1.9 ± 0.6 for Pd(Br) and 1.2 for Pd(Cl). I don't see an error value listed for the Pd(Cl) data. This is very important.
- I also see that morphologically authors captured parts of two individual particles in the case of Pd(Br) whereas in the case of Pd(Cl) authors show what looks like a 3 particle agglomerate in figure 4. The placement of the cylinders across the interfaces from where the Carbon concentration was measured to estimate the CTA+ excess can introduce substantial variability in this estimate which is an issue here especially due to the dissimilar morphology of the particles in both cases. So doing a one to one comparison of the CTA+ excess values in the manner presented will be extremely challenging. Authors however should provide (in supplementary info) all details on where the cylinders were placed and how the cylinder composition profiles looked along with the cumulative compositional plots appeared to further strengthen the interfacial excess calculations (as I mentioned in the overall comment above).
- Page 14, line 262, please correct: which well agrees with previous theoretical...

- The DFT calculations overall looks at the as-synthesized state. I wonder what DFT results have to say regarding how the high electric field applied during electrophoresis and electrodeposition might destabilize the binding between ligand and Pd (Cl) and Pd(Br) NP during the sample preparation process.

Reviewer #2 (Remarks to the Author):

The manuscript by Jang et al. proposes an interesting study on the crucial role of halides and cetrimonium (CTA) ions as surface active species directly involved in palladium nanoparticle growth and evolution in oxidative environment.

Using a multidisciplinary approach, which combined atom probe tomography (APT) and theoretical calculations (DFT), authors evidenced the complex scenario involved in the growth as well as in the dissolution behaviour of Pd NPs, since selective binding on crystal facets, competitive interactions between ligands and solvation effects can concomitantly determine the morphology of Pd NPs and their resistance to oxidative etching. Specifically, APT measurements revealed that a complete coverage of the metal surfaces by CTA is obtained only in the presence of bromide ions and this might account for the higher resistance to etching compared to analogous Pd NPs prepared in the presence of chloride ions. DFT calculations demonstrated that (a) the different adsorption behaviour could be ascribed to the denser packing of adsorbed bromide ions on Pd surface compared to chloride ions (with effects on the CTA adsorption and protection against etching) and (b) the presence of a great number of adsorption sites on Pd(111) facets could justify the preferential exposure of these facets (with effects on the shape-control).

The topic is highly debated and is of paramount importance for the design of synthetical strategies to obtain Pd and more generally metal nanoparticles with controlled shapes, sizes and physicochemical properties, for application in several fields, such as catalysis, optoelectronics and biomedicine.

The use of atom probe tomography represents a noteworthy element of novelty in so far as it allows a direct imaging of the spatial distribution of ligands at the metal surface.

In general, the paper is informative and contains potentially valuable results. However, I found it necessitates a revision before it could be accepted for publication on Nature Communications. Some comments and suggestions, that can help in improving the paper, are reported below:

- The collection of CO DRIFT spectra is highly recommended to reinforce the main conclusions made by the authors on the different packing density for Pd(Br) and Pd(Cl) and on the preferential adsorption on Pd(111) facets. Indeed, CO-DRIFTS is recognized as important technique for the characterization of Pd NPs, being CO adsorption sensitive to NP size, exposed facets and coverage dependent lateral interactions. Differently, the reported FT-IR spectra are less informative and could be moved to SI.
- Additionally, thermogravimetric techniques and X-ray photoelectron spectroscopy could provide useful information on the residual amount of capping agent on Pd NPs, to support the evidence from ATP. The authors could take in consideration also these measurements.
- p.7, line 137-139. Concerning long-term storage experiments, authors state that for a better comparison, KCl has been added to Pd(Br) sample, in order to realize the same concentration of chloride anions (etching agent) as in Pd(Cl). This approach is reasonable, however it should be also noted that the addition of KCl to Pd(Br) leads to an increase of salinity and ionic strength, which is not experienced in

the case of Pd(Cl). According to salting-out effect, the addition of electrolyte results in a decreased solubility of gases (specifically oxygen) in water and the decrease is proportional to the salt concentration and ionic strength. The decreased amount of dissolved oxygen might contribute to the observed attenuation of oxidative etching in the case of Pd(Br), could then authors add a comment in the text about this important point, please?

- PDOS calculations demonstrated that Pd forms stronger interactions with chloride anions than with bromide ones, as stated at p.14 lines 265-270. However, at p.19 lines 361-362 authors state that stronger bonds are formed between Pd and Br⁻. Can the author clarify this point, please? Is it correct to classify the interaction between Pd surface atoms and halide anions as covalent bonds?

- p.20, Synthesis of Pd nanoparticles. It is not declared which is the reducing agent for metal nanoparticle formation. Could the authors add this important information, please? Why did authors use CTAC also for the synthesis of Pd(Br)? CTAB (the cetrimonium bromide salt) is largely available and likely its use would have been preferable in the case of Pd(Br) NPs.

Minor points:

- Concerning the title, a more detailed version is suggested (e.g. "Three-dimensional Atomic Mapping of Ligands on Palladium Nanoparticles by atom probe tomography")

- The introduction is well written, it contains relevant references to the literature and clearly explains which are the main challenges in the field and the novelty of the work. I just suggest introducing more recent references on the role of halides in regulating the chemistry of metal nanoparticles (e.g. Nano Research, 2017, 10(3), 1064–1077; Nanoscale, 2019, 11, 15612–15621; Nanoscale, 2017, 9, 17914–17921; Nanoscale, 2016, 8, 3962–3972; PHYSICAL REVIEW MATERIALS, 2020, 4, 096001; ChemNanoMat 2020, 6, 576-588.).

Response to reviewers' comments

Reviewer #1

Authors attempt to characterize the halide ions and CTA+ ligands adsorbed on Pd nanoparticles using atom probe tomography and then use DFT calculations to validate the findings of lower CTA+ ligand coverage on Pd(Cl) versus Pd(Br). Then authors use both the APT and DFT results along with TEM results of air exposed nanoparticle stability to attribute the improved stability of Pd(Br) against oxidative etching on the higher coverage of CTA+ on Pd(Br) nanoparticles. While this work has some value as a first attempt to analyze organic ligands attached to nanoparticle by APT, I do have some serious concerns on the influence of sample preparation process on the entire APT experimental results presented here in this manuscript.

Comment 1:

“Specifically, once the CTA+ ligand coated Pd(Br) and Pd(Cl) nanoparticles are synthesized, they are then washed three times, centrifuged, then subjected to electrophoresis, then to electrodeposition to coat the Ni as a capping layer and then to FIB milling at room temperature to make the final APT needle. Shouldn't these steps influence the ligand distribution on nanoparticles rather substantially? These are rather intense mechanical, electrochemical, ion irradiation induced modification of the ligand layers before they are analyzed in APT. However, in this work authors show APT results from Pd(Br), Pd(Cl) and Pd nanoparticle free electrodeposited Ni coating. But I do feel authors have to show a result of clean Pd nanoparticles without Br or Cl or CTA+ but after that also went through the entire sample preparation process and show that there is no Carbonaceous species at the Ni-Pd nanoparticle interface to prove that the APT sample preparation is not what is leading to the carbon rich layer. Without this, directly attributing all the measured Carbon atoms to be coming only from CTA+ feels little bit like a leap of faith.”

Response 1:

We are thankful for these thoughtful comments and agree with the reviewer that the specimen preparation method is a critical part of this work. To address each of the concerns of the reviewer, we have subdivided our response as follows:

1-1. Influence of sample preparation steps on ligand distribution of nanoparticles

Influence of the washing process:

Indeed, the washing process not only removes the excess CTA⁺ molecules from the solution, but also some of the CTA⁺ molecules from the surfaces of the nanoparticle. Therefore, the absolute concentrations of ligands on the washed nanoparticles are lower than on the as-synthesized nanoparticles, as indicated by the FT-IR data (Supplementary Figure 4).

However, we want to emphasize that the objective of the atom probe (and DFT) studies was to relatively compare the Pd_(Br) and Pd_(Cl) specimens with respect to the absorption tendency and distribution of CTA⁺ molecules. The focus of this study was to elucidate how Br⁻ and Cl⁻ ions affect/promote CTA⁺ adsorption. To this end, washed nanoparticles can be used. In order to make a direct comparison between Pd_(Br) and Pd_(Cl) specimens, we made sure that both specimens were washed under the same conditions, *i.e.*, using identical time, speed and volume of solvent for centrifuging.

Influence of the electrophoresis and electrodeposition process:

For the electrophoresis and electrodeposition process, we used a vertical cell, which is described in a previous paper (Kim et al. *Ultramicroscopy* 190 (2018): 30-38). A schematic of this cell is shown in Fig. R1-1. During the deposition process, Pd nanoparticles and Ni ions of positive surface charge migrate in the solution towards a Cu substrate (cathode) at negative bias (~ -4 V) due to electrostatic forces and gravity, and are deposited as a composite film on the electrode by a reduction process. Since the used Cu substrate is a planar foil of (~ 0.2 μm surface roughness) and thus shows a much lower curvature than the Pd nanoparticles, the surface charge density of the nanoparticles should be substantially higher than that of the substrate. Thus, the nanoparticles should exert a much stronger attractive Coulomb force on the ligand ions than the planar Cu substrate and the ligand ions are not expected to become detached from the nanoparticles during the electrophoresis and electrodeposition process.

Figure R1-1. A schematic of the vertical cell used for electrophoresis and electrodeposition process. **a, b** electrophoresis of NPs on the cathode (Cu substrate). **c, d** electrodeposition of a Ni layer on the cathode. (Reproduced from Kim et al. *Ultramicroscopy* 190 (2018): 30-38)

Furthermore, the electrodeposition process itself only introduces a negligible amount of impurities (including C), as shown in our experiments on bare electrodeposited Ni films (see supplementary material, Supplementary Fig. 7) and our experiments on “clean” Pd nanoparticles (see reply 1-2 below).

Influence of the FIB process:

In order to reduce the implantation of Ga^+ ions during FIB milling to a minimum level, we applied a final low-kV (5 kV) clean-up step for the sharpened APT specimens. Also, the concentrations of implanted Ga^+ ions were determined from the APT mass spectra. We extracted cuboidal ROIs ($40 \times 40 \times 40 \text{ nm}^3$ in size) containing Pd nanoparticles and surrounding ligands from the APT datasets of $\text{Pd}_{(\text{Br})}$ and $\text{Pd}_{(\text{Cl})}$ (Supplementary Fig. 10). The peaks at 34.5 Da could be assigned to Ga^{2+} , whereas the peak at 69 Da could be assigned to Ga^+ or Ni_2O^+ ions. We also note that monoisotopic ^{69}Ga is used for the FIB process, and therefore the peaks at 35.5 and 71 Da could be safely assigned to elements other than Ga. When assigning the peaks at 34.5 and 69 Da to Ga ions, the resulting concentrations were 0.02 and 0.07 at.% for the $\text{Pd}_{(\text{Br})}$ and $\text{Pd}_{(\text{Cl})}$ specimens, respectively. However, the peaks in the 66 – 72 Da region could be assigned to Ni_2O^+ , indicating that a major portion of the peak at 69 Da is due to Ni_2O^+ ions. Therefore, the actual Ga concentrations are expected to be substantially lower than the upper values of 0.02 and 0.07 at.% for $\text{Pd}_{(\text{Br})}$ and $\text{Pd}_{(\text{Cl})}$, respectively. Thus, we conclude that Ga ion induced damage is largely removed during the low-kV clean-up process, and that artifacts caused by Ga^+ implantation during FIB milling is negligible.

To clarify the Ga implantation issue, we added the following sentence to the “Methods – TEM and APT characterization” section (page 23, line 456):

“In order to reduce the implantation of Ga⁺ ions during FIB milling to a minimum level, we applied a final low-kV (5 kV) clean-up step for the sharpened APT specimens.”

Also, we added the following sentence to the “Results - 3D distribution and concentration of CTA⁺ ligands on the Pd NPs” section (page 10, line 201):

“Additionally, the level of implanted Ga⁺ ions was estimated by analyzing the bulk mass spectra of cuboidal regions of interest (ROIs) (40×40×40 nm³ in size) containing the Pd NPs and surrounding ligands. Taking into account the peak overlap between the Ga⁺ and Ni₂O⁺ ions at 69 Da (see Supplementary Fig. 10), the maximum Ga concentrations around the NPs in the Pd_(Br) and Pd_(Cl) specimens were estimated to be 0.02 and 0.07 at.%, respectively.”

And we added Supplementary Figure 10 to Supporting information.

Supplementary Figure 10. Mass spectra acquired from ROIs (40×40×40 nm³ in size) containing Pd NPs and surrounding ligands; Pd_(Br) (top) and Pd_(Cl) (bottom). For evaluating the upper concentration limits of the implanted Ga ions, the mass spectra are only shown in the ranges from 28.5 to 36.5 Da and from 56.5 to 72.5 Da.

1-2. Additional APT experiments on “clean” Pd nanoparticles

In order to prepare nanoparticles showing “clean” surfaces free of carbonaceous species, as suggested by the reviewer, we had to change the synthesis route. We used K_2PdCl_4 and NaBH_4 as a precursor and a reductant, respectively, and did not add any CTA^+ . Thus, no C-containing chemical was used in this recipe. The obtained Pd nanoparticles are shown in Figure R1-2a. They show irregular shape, as no shape-controlling agent was used.

We used these nanoparticles for preparing APT specimens under identical conditions as the $\text{Pd}_{(\text{Br})}$ and $\text{Pd}_{(\text{Cl})}$ nanoparticles (including washing, electrophoresis and Ni electrodeposition, FIB milling). The resulting APT data is shown in Figure R1-2b. In contrast to the $\text{Pd}_{(\text{Br})}$ and $\text{Pd}_{(\text{Cl})}$ nanoparticle specimens, no remarkable C-enrichments can be seen in the atom map. From the overall mass spectrum derived from the atom map (Figure R1-3), the C content was determined to be about 35 ppm. Moreover, no carbon segregation was detected on the Pd surface.

Thus, we can clearly conclude that the C segregation zones shown in our APT data in Fig. 2 and Supplementary Fig. 11 indeed originate from the CTA^+ ions used in the synthesis process and is not an artifact of APT sample preparation process.

Figure R1-2. TEM and APT analysis of Pd NPs synthesized without CTA^+ . **a** Bright field TEM image. **b** three-dimensional atom map of the NPs with iso-concentration (50 at.%) surfaces.

Figure R1-3. Mass spectrum acquired from a Pd NP specimen, prepared using K_2PdCl_4 and NaBH_4 but not using CTA^+ . Electrophoresis, electrodeposition, and FIB milling processes were identical to those applied to the $\text{Pd}_{(\text{Br})}$ and $\text{Pd}_{(\text{Cl})}$ specimens.

We added the following text in the manuscript and Supplementary Fig. 8 to supplementary information.

“To further clarify the origin of the C and N atoms detected in the $\text{Pd}_{(\text{Br})}$ and $\text{Pd}_{(\text{Cl})}$ specimens, we prepared Pd NPs, using K_2PdCl_4 and NaBH_4 as a precursor and a reductant, respectively, and not using any CTA^+ ligands. APT specimens were prepared from these NPs, using the same procedure as for the $\text{Pd}_{(\text{Br})}$ and $\text{Pd}_{(\text{Cl})}$ specimens. As can be seen in the acquired APT data (see Supplementary Figure 8) the detected C concentration was diminishingly low (~ 35 ppm) as compared to the $\text{Pd}_{(\text{Br})}$ and $\text{Pd}_{(\text{Cl})}$ specimens. Moreover, no significant C segregation was detected on the Pd surface. Thus, we conclude that the C segregation zones detected for the $\text{Pd}_{(\text{Br})}$ and $\text{Pd}_{(\text{Cl})}$ specimens can be indeed ascribed to the CTA^+ ions.” (page 9, line 188)

Supplementary Figure 8. APT data acquired from a Pd NP specimen, prepared using K_2PdCl_4 and NaBH_4 but not using CTA^+ . Electrophoresis, electrodeposition, and FIB milling processes were identical to those applied to the $\text{Pd}_{(\text{Br})}$ and $\text{Pd}_{(\text{Cl})}$ specimens. (Scale bar: 10 nm)

Comment 2:

“Also authors have to show all the details of the interfacial excess calculation from APT because that’s one of the main point highlighted in this paper. Also the error value for the CTA⁺ excess in Pd(Cl) needs to be provided. I also have concern on how repeatable these carbon excess quantification are from particle to particles. So rather than just providing the average of carbon excess for 8 measurements, authors should provide a scatter plot with error bars to show is there a clear above error bar level difference in the measured carbon excess between Pd(Br) and Pd(Cl) to be conclusive in the results. Without that this result may not be repeatable.”

Response 2:

Thank you for pointing out this suggestion. We included an additional section in the supplementary information, describing the details on the determination of the interfacial excess value for an individual nanoparticle. Furthermore, we listed the individual excess values in a separate table (Supplementary Table 2) and in a scatter plot (Supplementary Fig. 13), as requested by the reviewer. The added section in the supplementary information is as follows.

“Determination of the Gibbsian interfacial excess

In order to compare the amount of the segregated CTA⁺ ions on each Pd sample, the C excess was determined using the method proposed by Krakauer et al.

First, a cylindrical ROI was placed perpendicular across the interface between a Pd NP and Ni matrix. Next, the cumulative number of C atoms was plotted against the cumulative total number of atoms along the z-direction of the cylindrical ROI. Such a cumulative plot is also termed ladder diagram. From the ladder diagram, the total number of C atoms segregated on the Pd surface (C_{excess}) was obtained by subtracting the cumulative number of C atoms in the Ni matrix (C_{min}) from the cumulative number of C atoms in the Ni matrix and the interface (C_{max}). A linear function was fitted to the linear section of cumulative curve in the Ni matrix region to obtain the value of C_{min} . Similarly, another linear function was fitted to the linear section of cumulative curve in the Pd NP region to yield the value of C_{max} . In the cases where the linear plateau was not observed in the cumulative plot of the Pd NP region, the total number of C atoms were regarded as C_{max} , as done in Ref^{1,2}.

The measured C_{excess} value can be used for the calculation of the surface density of CTA⁺ ions by using equation 1 in the manuscript. The calculation of the interfacial excess and the surface density of CTA⁺ ions for one exemplary ROI across a Pd(Br) NP (ROI #5 in Supplementary Table 2) is as follows:

$$\Gamma_C = \frac{1}{A\eta} N_C^{excess} = \frac{1}{19.6 \text{ nm}^2 \times 0.37} (268 \text{ C atoms}) = 36.9 \frac{\text{C atoms}}{\text{nm}^2} \triangleq 1.9 \frac{\text{CTA molecules}}{\text{nm}^2}$$

The C interfacial excess is 36.9 C atoms/nm² for the given ROI. Since there are 19 C atoms in one CTA molecule, the surface density of the molecules is 1.9 CTA molecules/nm². For each Pd_(Br) NP, four ROIs were placed across the interface between the Pd NP and the matrix. For the Pd_(Cl) NPs, eight ROIs were randomly positioned across the interface. The determined average values of the CTA⁺ surface density were 1.9 ± 0.2 and 0.7 ± 0.3 CTA⁺ / nm² for the Pd_(Br) and the Pd_(Cl) NPs, respectively. While the data points are more scattered in the case of Pd_(Cl), the difference in the CTA⁺ surface density between Pd_(Br) and Pd_(Cl) is clearly beyond the error range (See Supplementary Table 2 and Supplementary Figure 13).

Supplementary Table 2. Surface density values of CTA⁺ ions adsorbed on Pd_(Br) and Pd_(Cl) specimens, as determined from ladder diagrams.

ROI	CTA ⁺ /nm ² Pd _(Br)	CTA ⁺ /nm ² Pd _(Cl)
#1	1.8	0.7
#2	1.6	0.7
#3	1.8	0.8
#4	2.2	0.2
#5	1.9	0.3
#6	1.7	0.8
#7	2.1	1.2
#8	1.7	1.1
Average	1.9 ± 0.2	0.7 ± 0.3

Supplementary Figure 13. A scatter plot of the surface density of CTA⁺ ions on the Pd_(Br) (orange) and Pd_(Cl) (purple) specimens. Individual values are listed in Supplementary Table 2, where the average values are 1.9 ± 0.2 and 0.7 ± 0.3 CTA⁺/nm² for Pd_(Br) and Pd_(Cl), respectively. (Error bars represent a standard deviation and × represents an average.)

Supplementary References

1. Hellman, O. C. & Seidman, D. N. Measurement of the Gibbsian interfacial excess of solute at an interface of arbitrary geometry using three-dimensional atom probe microscopy. *Mater. Sci. Eng. A* **327**, 24–28 (2002).
2. Hellman, Vandembroucke, Rusing, Isheim & Seidman. Analysis of Three-dimensional Atom-probe Data by the Proximity Histogram. *Microsc. Microanal.* **6**, 437–444 (2000). ”

We have also added the following text in the manuscript, at page 13, line 253.

“For both Pd_(Br) and Pd_(Cl) specimens, eight cylindrical ROIs were placed at different locations across the interface between a NP and the matrix. The corresponding C excess values were determined from a cumulative plot of the C atoms against the total number of atoms within the ROI according to the method proposed by Krakauer et al⁴⁹. Further details on this method are given in the supporting information, where the results are listed in Supplementary Table 2 and shown as a scatter plot in Supplementary Fig. 13. The determined average values of the cetrimonium surface density were 1.9 ± 0.2 and 0.7 ± 0.3 CTA⁺ / nm² for the Pd_(Br) and the Pd_(Cl) NPs, respectively. While the data points are more scattered in the case of Pd_(Cl), the difference in the CTA⁺ surface density between Pd_(Br) and Pd_(Cl) is clearly beyond the error range (See Supplementary Table 2 and Supplementary Figure 13).”

Comment 3:

“The second statement in abstract “interaction between capping ligands and their interaction with the crystallographic surfaces of nanoparticles during colloidal synthesis remains a great mystery, due to the lack of direct imaging techniques.” is a bit of an overstatement. There are great many spectroscopic techniques to analyze ligand interaction.”

Response 3:

We agree with the reviewer’s suggestion. We revised the sentence in the abstract as follows.

“However, the synergistic effect between different ligands and their distribution on the crystallographic surfaces of nanoparticles during colloidal synthesis is still unclear despite the powerful spectroscopic techniques, due to a lack of direct imaging techniques.” (page 2, line 21)

Comment 4:

“Please provide the chemical formula of cetrimonium in the abstract.”

Response 4:

Thank you for this comment. We added the chemical formula of cetrimonium in the abstract.

Comment 5:

“The FTIR figure doesn’t seem crucial for the manuscript and hence can be moved to supplementary information.”

Response 5:

We moved the FT-IR data to supplementary data as Supplementary Figure 4.

Supplementary Figure 4. Fourier-transform infrared (FT-IR) spectra. Pure cetrimonium ions (CTA⁺) and Cl anions, compared with Pd NPs produced by adding Br or Cl anions (Pd_(Br) and Pd_(Cl), respectively), as-synthesized and after washing to remove excess CTA⁺.

Comment 6:

“In supplementary figures 3 and 4 APT mass-to-charge spectra, the y axis labels should be log (counts)?”

Response 6:

We agree with the reviewer. We changed the figure axis to log (counts) (see Supplementary Fig. 5 and 6).

Comment 7:

“I notice the spectra FWHM is deteriorated considerably in the case of Pd(Cl) versus Pd(Br) why is this? And how is this influencing the interpretations given in this work?”

Response 7:

We thank the reviewer for this remark. Since the mass spectra of the Pd_(Cl) and Pd_(Br) specimens were originally shown with different bin widths (0.102 Da and 0.051 Da, respectively), we changed the bin width of the mass spectra of Pd_(Cl) to the same value of those of Pd_(Br) for proper comparison. Supplementary Fig. 5 and 6 shows the revised mass spectra of Pd_(Br) and Pd_(Cl), indicating that the difference in FWHM of the Pd peaks (104, 105, 106, 108, and 110 Da) between the two specimens is not significantly different (average FWHM of the Pd peaks: 0.202 Da for Pd_(Br) and 0.206 Da for Pd_(Cl)). We also want to emphasize that the mass spectra are shown in logarithmic scale, and therefore the noise levels appear much higher than they are in linear scale.

More pronounced “thermal tails”, arising from laser pulsing, can be seen for the Pd_(Cl) specimen. However, these are pronounced for the non-relevant Ni-related peaks and are less visible for the relevant C-related peaks. The width of a thermal tail can depend on various parameters, such as the laser pulse frequency, pulse energy, specimen base temperature, and the geometry of the tip (Bunton et al. *Microsc. Microanal.* 13.6 (2007): 418). In our case, we tried to minimize the effects of these parameters by keeping the measurement conditions as similar as possible. However, the specimen geometry is very difficult to control with high precision due to the FIB milling process, which slightly varies from tip to tip. Thus, slight differences in tip geometry may have led to different cooling rates during cooling and the observed differences in the “thermal tails”.

Since defining the width of each mass peak has a significant effect on the sample composition, we applied the same peak ranging method (ranging at full-width of tenth maximum) to both specimens to properly compare the APT data.

Supplementary Figure 5. APT mass spectrum acquired from Pd_(Br) NPs embedded in Ni. The spectrum is shown for different mass-to-charge ratio ranges in the top, middle, and bottom figure.

Supplementary Figure 6. APT mass spectrum acquired from Pd_(Cl) NPs embedded in Ni. The spectrum is shown for different mass-to-charge ratio ranges in the top, middle, and bottom figure.

Comment 8:

“Authors needs to show an overlaid comparison of Cl peaks and peaks at the locations where Br peaks should be present between the two samples. Given the high likelihood of substantial overlaps between organic peaks and Cl peaks as well as Br peaks there can be considerable uncertainty in assigning the peaks at 35 and 37 Da as well as peaks at 39.5 and 40.5 Da peaks.”

Response 8:

Thank you for this thoughtful comment. We indicated the peak positions of the halide ions in the mass spectra of both specimens (see Supplementary Fig. 9. Br peaks were not indicated in the Pd_(Cl) specimen, since no Br-containing chemical was used for synthesizing Pd_(Cl). We added Supplementary Figure 9 to supplementary information.

Supplementary Figure 9. Selected parts of the mass spectra of a Pd_(Br) (left) and Pd_(Cl) (right) specimen for closer inspection of the Br and Cl peaks. The light purple and red bars in the mass spectrum indicate the positions of the Br and Cl peaks, respectively, where their heights indicate the natural isotope ratios.

We also added following paragraphs to supplementary information:

“Supplementary Figure 9 shows a comparison of the mass spectra ranges of both Pd_(Br) (left) and Pd_(Cl) (right) specimens, where the expected positions of the Cl⁺, Cl²⁺, Br⁺, and Br²⁺ peaks are marked by bars. For the Pd_(Br) specimen, Br²⁺ peaks expected at 39.5 and 40.5 Da are below the detection limit. Slight peaks are detected at 79 and 81 Da, which can be partly assigned to Br⁺ ions. Assigning these two peaks to Br⁺ gives an upper limit for the Br concentration in the given dataset, which is 0.008 at.%. However, it is more reasonable to assign the peaks at 79 and 81 Da to ⁶²NiOH⁺ and ⁶⁴NiOH⁺ molecular ions, since the heights of the peaks detected in the range from 75 to 81 Da range show a good match with the natural abundance of Ni isotopes. Thus, the actual Br concentration is expected to be substantially lower than 0.008 at.%.

For both Pd_(Br) and Pd_(Cl) specimens, mass peaks at 35 and 37 Da were detected. Assigning these peaks to Cl⁺ yields upper limits of the Cl concentrations in the given datasets, which are 0.024 at.% for Pd_(Br) and 0.069 at.% for Pd_(Cl). However, the measured ratios between the peaks at 35 and 37 Da show deviations from the natural isotope ratio (1.36:1), with values of 1.08:1 for Pd_(Br) and 1.04:1 for Pd_(Cl). These results are indicative of the actual Cl concentrations being substantially lower than the determined upper limits.”

And we modified the manuscript as follow:

“The detected halide ion concentrations were very low in both Pd_(Br) and Pd_(Cl) specimens, namely < 0.008 at% Br in Pd_(Br), < 0.024 at% Cl in Pd_(Br), and < 0.069 at% Cl in Pd_(Cl) (See supplementary Fig. 9).” (page 10, line 196)

Comment 9:

“About Figure 3a and b authors talk about pentagonal projection and say this is a decahedral particle. Its difficult to see conclusively from the APT figures.”

Response 9:

We agree with the reviewer’s comment. We have revised the manuscript accordingly (page 10, line 209) and changed the sentence from

“The 10 nm thin slice viewed along the z-axis shown in Fig. 3b reveals a nearly pentagonal projection for the top Pd NP, suggesting that it is an MTNP with a decahedral shape.” to

“The 10 nm thin slice viewed along the z-axis shown in Fig. 2b reveals a projection of the top Pd NP.”

Comment 10:

“In figure 3 it is recommended to show how Cl is distributed in the Pd(Br) samples also, since authors are showing evidence for clear peaks of Cl detected in the APT mass to charge spectra of Pd(Br) in supplementary figure 3(a) but it is missing in figure 3 of manuscript.”

Response 10:

We are grateful for the reviewer’s suggestion. We assigned the peaks at 35 and 37 Da to Cl ions and added the Cl distribution to Fig. 2.

Fig. 2 Atom probe tomography (APT) reconstruction of Pd multiple-twinned NPs (MTNPs), synthesized by adding Br anions, embedded in Ni. a Three-dimensional atom map of Pd atoms and iso-concentration (1.5 at.%) surfaces of C. **b, c** Slices viewed along the Pd MTNPs. (Scale bars: 10 nm)

Comment 11:

“In page 10 authors mention the C:N ratio was 1:18.4. To support this, it is recommended to show the cylinder line composition profile perpendicular to the Pd-Ni interface from where this was estimated in the supplementary data with error bars.”

Response 11:

Local cylinder line composition profiles did not give any reliable values for the C:N ratio, because the number of atoms (in particular nitrogen) was too small. Originally, we measured the C:N ratio using bulk mass spectrum analyses of small ROIs ($15 \times 15 \times 15 \text{ nm}^3$ in size) containing C-complexes on the surfaces of Pd NPs. To improve the reliability of the C:N ratio, we determined the C:N ratios from the bulk compositions of large cubical ROIs (see Supplementary Fig. 12) which completely covered the surface areas of the NPs. These ROIs also contained significant portions of the Ni matrix, in order to include CTA⁺ molecules expanding from the Pd NPs into the Ni matrix. We want to emphasize here once again that C impurities due to the electrodeposition process can be neglected (see response 1-1 and 1-2).

To confirm our previous finding, we determined the C:N ratio from three different sampled regions (see Supplementary Fig. 12) of the Pd_(Br) sample. We revised the following text in the manuscript in page 10, line 215.

“The average bulk compositions of three different regions containing C-complexes (marked by the C iso-concentration surface) showed a C:N ratio of 18.9:1 (see Supplementary Fig. 12 and Supplementary Table 1), confirming that the C and N atoms indeed originated from the CTA⁺ ions. Thus, these APT results indicate the segregation of CTA⁺ on the surface of the Pd_(Br) NPs.”

Also, we have included Supplementary Fig. 12 and Supplementary Table 1 in the supplementary material.

Supplementary Figure 12. APT reconstructions of Pd_(Br) MTNPs and cubical ROIs from which the C:N ratios were determined.

Supplementary Table 1. Number of C, N atoms and C:N ratio of each ROI shown in Supplementary Figure 12.

	C counts	N counts	C:N ratio
ROI #1	6192	344	18.0
ROI #2	1945	128	15.2
ROI #3	17221	868	19.8
Total	25358	1340	18.9

Comment 12:

“Unclear why authors used 1.5 at% C isoconcentration surface in Pd(Br) in figure 3(a) and 3 at% isoconcentration surface in the case of Pd(Cl) in Figure 4(a)?”

Response 12:

The C iso-concentration surfaces just serve as a means to properly visualize the C-rich regions in the corresponding datasets. For the Pd_(Br) specimen, a thresh-hold value of 1.5 at.% C yielded the most illustrative visualization, whereas 1.5 at.% C was not the optimum thresh-hold value for visualizing the C enrichments in the Pd_(Cl) atom map, as the iso-surfaces covered almost the entire atom map (see Figure R12d). In contrast, when choosing a concentration threshold above 3 at.%, C appeared to be located only at the top of the atom map, although C atoms were also found around the NPs in the atom map. Therefore, we chose 3 at.% for the Pd_(Cl) specimen.

Figure R12. Iso-concentration surfaces of C (blue) with varying concentrations in the atom maps of Pd_(Br) and Pd_(Cl). a–c Pd_(Br) atom maps with iso-concentration C surfaces of 0.5, 1.5, 2.5 at.%, respectively. d–f Pd_(Cl) atom maps with iso-concentration C surfaces of 1.5, 3.0, 4.5 at.%, respectively. Blue dots indicate carbon atoms.

Comment 13:

“The most highlighted finding in this paper is the interfacial excess of CTA⁺ of 1.9 ± 0.6 for Pd(Br) and 1.2 for Pd(Cl). I don’t see an error value listed for the Pd(Cl) data. This is very important.”

Response 13:

We have added a section about the determination of the surface density of the CTA⁺ ions to supplementary information (see response 2). Accordingly, we have included error values for the Pd_(Cl) specimen in the manuscript in page 13, line 260.

Comment 14:

“I also see that morphologically authors captured parts of two individual particles in the case of Pd(Br) whereas in the case of Pd(Cl) authors show what looks like a 3 particle agglomerate in figure 4. The placement of the cylinders across the interfaces from where the Carbon concentration was measured to estimate the CTA⁺ excess can introduce substantial variability in this estimate which is an issue here especially due to the dissimilar morphology of the particles in both cases. So doing a one to one comparison of the CTA⁺ excess values in the manner presented will be extremely challenging. Authors however should provide (in supplementary info) all details on where the cylinders were placed and how the cylinder composition profiles looked along with the cumulative compositional plots appeared to further strengthen the interfacial excess calculations (as I mentioned in the overall comment above).”

Response 14:

We thank the reviewer for this thoughtful comment. Indeed, it was difficult to clearly distinguish between individual Pd_(Cl) nanoparticles in the atom map of the three particle agglomerate. Thus, eight cylindrical ROIs were placed randomly across the interface between Ni and Pd_(Cl). The method for determining the CTA⁺ excess values was already described in reply 2 and details are given in the supplementary Information.

We have added the following text to page 13, line 253.

“For both Pd_(Br) and Pd_(Cl) specimens, eight cylindrical ROIs were placed at different locations across the interface between a NP and the matrix. The corresponding C excess values were determined from a cumulative plot of the C atoms against the total number of atoms within the ROI according to the method proposed by Krakauer et al⁴⁹. Further details on this method are given in the supporting information, where the results are listed in Supplementary Table 2 and shown as a scatter plot in Supplementary Fig. 13. The determined average values of the cetrimonium surface density were 1.9 ± 0.2 and 0.7 ± 0.3 CTA⁺ / nm² for the Pd_(Br) and the Pd_(Cl) NPs, respectively. While the data points are more scattered in the case of Pd_(Cl), the difference in the CTA⁺ surface density between Pd_(Br) and Pd_(Cl) is clearly beyond the error range (see Supplementary Table 2 and Supplementary Figure 13).”

Comment 15:

“Page 14, line 262, please correct: which well agrees with previous theoretical...”

Response 15:

We corrected the sentence to: “which is in good agreement with previous theoretical...” at page 14, line 291.

Comment 16:

“The DFT calculations overall looks at the as-synthesized state. I wonder what DFT results have to say regarding how the high electric field applied during electrophoresis and electrodeposition might destabilize the binding between ligand and Pd(Cl) and Pd(Br) NP during the sample preparation process.”

Response 16:

Thank you for this excellent comment.

We are well aware that the binding strength of the ligands on the Pd NPs is likely to be altered by the electric field applied during the electrophoresis and electroplating processes. Although evaluating the change in ligand binding strength under an electric field is of great scientific interest, we did not make a DFT based prediction of such a phenomenon for the following reasons:

Since the halide-ligand pairs cover the NP surfaces in three dimensions and the as-synthesized non-spherical NPs are randomly oriented during electrodeposition, an electric field is likely to induce anisotropic variations in the ligand-binding strengths depending on the orientation of the facets (see Figure R16). In this regard, it seems impractical to draw certain conclusions based on DFT-based predictions of ligand-binding strength for selected facets under a given electric field.

Moreover, we want to emphasize that DFT calculations are hindered by the challenge of considering a finite electric field under periodic boundary conditions (Fu, C. L. and K. M. Ho. Phys. Rev. Lett. 63.15 (1989): 1617). Recently, a state-of-the-art DFT algorithm to overcome this issue has been proposed (Ashton et al. Phys. Rev. Lett. 124.17 (2020): 176801), but the implementation of this algorithm is limited to the developer’s in-house DFT code and needs further validation. Based on these recent developments in the field of DFT, we envision to address the effects of an electric field on the ligand-binding strength in a future study. However for this work, such calculations are out of scope.

Figure R16. Schematic illustration of nanoparticles under an electric field.

Reviewer #2

The manuscript by Jang et al. proposes an interesting study on the crucial role of halides and cetrimonium (CTA) ions as surface active species directly involved in palladium nanoparticle growth and evolution in oxidative environment.

Using a multidisciplinary approach, which combined atom probe tomography (APT) and theoretical calculations (DFT), authors evidenced the complex scenario involved in the growth as well as in the dissolution behaviour of Pd NPs, since selective binding on crystal facets, competitive interactions between ligands and solvation effects can concomitantly determine the morphology of Pd NPs and their resistance to oxidative etching. Specifically, APT measurements revealed that a complete coverage of the metal surfaces by CTA is obtained only in the presence of bromide ions and this might account for the higher resistance to etching compared to analogous Pd NPs prepared in the presence of chloride ions. DFT calculations demonstrated that (a) the different adsorption behaviour could be ascribed to the denser packing of adsorbed bromide ions on Pd surface compared to chloride ions (with effects on the CTA adsorption and protection against etching) and (b) the presence of a great number of adsorption sites on Pd(111) facets could justify the preferential exposure of these facets (with effects on the shape-control).

The topic is highly debated and is of paramount importance for the design of synthetical strategies to obtain Pd and more generally metal nanoparticles with controlled shapes, sizes and physicochemical properties, for application in several fields, such as catalysis, optoelectronics and biomedicine.

The use of atom probe tomography represents a noteworthy element of novelty in so far as it allows a direct imaging of the spatial distribution of ligands at the metal surface.

In general, the paper is informative and contains potentially valuable results. However, I found it necessitates a revision before it could be accepted for publication on Nature Communications. Some comments and suggestions, that can help in improving the paper, are reported below:

Comment 1:

“The collection of CO DRIFT spectra is highly recommended to reinforce the main conclusions made by the authors on the different packing density for Pd(Br) and Pd(Cl) and on the preferential adsorption on Pd(111) facets. Indeed, CO-DRIFTS is recognized as important technique for the characterization of Pd NPs, being CO adsorption sensitive to NP size, exposed facets and coverage dependent lateral interactions.”

Response 1:

We thank the reviewer for this comment. We performed CO-DRIFT measurements on washed Pd_(Br) and Pd_(Cl) NPs to support the APT and DFT results. We added the sentence below to the manuscript and added Supplementary Figure 16 and Supplementary Table 4 with the corresponding references to supplementary information.

In manuscript

“Furthermore, these results were also supported by CO-DRIFT measurements performed on washed Pd_(Br) and Pd_(Cl) NPs, where more CO molecules were adsorbed on the Pd_(Cl) NPs than on the Pd_(Br) NPs due to a lower amount of adsorbed CTA⁺ (Supplementary Fig. 16 and Supplementary Table 4).” (page 14, line 266)

“**CO-DRIFT measurements.** A Nicolet iS 50 FT-IR spectrometer (Thermo Scientific, USA) equipped with an in-situ cell was used for CO-DRIFT measurements at room temperature. NPs dried at 50 °C for 12 h were pretreated at room temperature in a He environment for 1 h to record a background spectrum. Subsequently, a mixture of He gas and 1 vol.% of CO was injected into the cell at a flow rate of 30 ml min⁻¹ until saturation. After purging with nitrogen (60 ml min⁻¹) to remove the physisorbed CO molecules, the DRIFT spectra were recorded at a resolution of 4 cm⁻¹.” (page 23, line 443)

In supporting information

Supplementary Figure 16. Acquired (dashed lines) and deconvoluted (colored areas) CO-DRIFT spectra of washed Pd_(Br) and Pd_(Cl) NPs. Peaks highlighted in pink and purple can be assigned to bridged CO bonds (2000 – 1700 cm⁻¹) and linear CO bonds (2100 – 2000 cm⁻¹) on the Pd surface, respectively. Peaks at 2121 and 2128 cm⁻¹ highlighted in green indicate linear CO bonds on Pd ions.

Supplementary Table 4. Wavenumber range and the assignment of each peak in the CO-DRIFT spectra of the washed Pd_(Br) and Pd_(Cl) NPs⁷⁻⁹.

Wavenumber (cm ⁻¹)	Assignment	Ref
2128, 2121	Linear CO on Pd ions	[7]
2099	Linear CO on Pd {100} facets	[7]
2063 – 2052	Linear CO on Pd defects	[7]
2032, 2018, 2004	Linear CO on Pd sites / Shifted linear CO on Pd defects	[8,9]
1912 – 1836	Threefold bridged CO on Pd {111} facets	[7]
1794	Fourfold bridged CO on Pd {111} facets	[7]

Supplementary Table 4 shows the wavenumber range and the assignment of each peak shown in Supplementary Fig. 16. Compared to the CO-DRIFT spectrum of the washed Pd_(Cl) specimen, the spectrum of the washed Pd_(Br) specimen clearly showed lower peak intensities, especially in the 1912 – 1836 cm⁻¹ region, indicating less CO adsorption on the {111} facets of Pd_(Br) compared to Pd_(Cl). This trend can be explained by the fact that the {111} sites of Pd_(Br) were covered with a higher density of CTA⁺ ligands than those of Pd_(Cl) in accord with the APT and DFT results¹⁰.

Supplementary References

7. Tereshchenko, A. et al. Pd nanoparticle growth monitored by DRIFT spectroscopy of adsorbed CO. *Analyst* 145, 7534–7540 (2020).
8. Zhang, L. et al. Efficient and durable Au alloyed Pd single-atom catalyst for the Ullmann reaction of aryl chlorides in water. *ACS Catal.* 4, 1546–1553 (2014).
9. Zeinalipour-Yazdi, C. D., Willock, D. J., Thomas, L., Wilson, K. & Lee, A. F. CO adsorption over Pd nanoparticles: A general framework for IR simulations on nanoparticles. *Surf. Sci.* 646, 210–220 (2016).
10. McKenna, F. M. & Anderson, J. A. Selectivity enhancement in acetylene hydrogenation over diphenyl sulphide-modified Pd/TiO₂ catalysts. *J. Catal.* 281, 231–240 (2011).

Comment 2:

“Differently, the reported FT-IR spectra are less informative and could be moved to SI.”

Response 2:

Thank you for your suggestion. We moved the FT-IR data to supplementary information as Supplementary Figure 4.

Comment 3:

“Additionally, thermogravimetric techniques and X-ray photoelectron spectroscopy could provide useful information on the residual amount of capping agent on Pd NPs, to support the evidence from APT. The authors could take in consideration also these measurements.”

Response 3:

Thank you for these great suggestions.

We performed thermogravimetric (TGA) and X-ray photoelectron spectroscopy (XPS) analyses on washed Pd_(Br) and Pd_(Cl) NPs to support the APT results. We added the sentence below to the manuscript and added Supplementary Figure 14, 15, and Supplementary Table 3 with some explanatory texts and references to supplementary information.

In manuscript

“The acquired APT results could be qualitatively confirmed by thermogravimetric analyses (TGA) and X-ray photoelectron spectroscopy (XPS) on washed Pd_(Br) and Pd_(Cl) NPs (see Supplementary Fig. 14, 15, and Supplementary Table 3).” (page 13, line 263)

Thermogravimetric analysis (TGA). A TG209 F1 Libra (NETZSCH) instrument was used for TGA measurements in the temperature range from 50 to 700 °C at a heating rate of 10 °C/min. All measurements were carried out in a nitrogen atmosphere. (page 23, line 434)

X-ray photoelectron spectroscopy (XPS) analysis. A Kratos Axis-Supra instrument was used for XPS measurements using monochromatic Al K α radiation (1486.7 eV). Photoelectrons were collected at a take-off angle of 90° relative to the sample surface. Data analyses were performed using the Thermo Scientific Avantage software. The binding energies of the spectra were calibrated by setting the C–C binding energy of the C1s peak to 284.8 eV. (page 23, line 437)

In supplementary Information

Supplementary Figure 14. TGA measurements. Weight loss curves given as wt.% vs. temperature (solid lines) and weight loss rate curves given as wt.%/min vs. temperature (dotted green lines) of **a** pure CTAC, **b** washed Pd_(Br) NPs, **c** washed Pd_(Cl) NPs.

Supplementary Figure 14 shows the TGA curves for pure CTAC, washed Pd_(Br), and washed Pd_(Cl) specimens. The TGA curve of CTAC (Supplementary Fig. 14a) exhibits only one weight loss step in the range from 200 to 260 °C, indicating the thermal decomposition of CTAC molecules³. By contrast, the curves of washed Pd_(Br) and Pd_(Cl) NPs (Supplementary Fig. 14b, c) show three weight loss steps in the range from 200 to 275 °C, from 275 to 345 °C, and from 345 to 450 °C, as seen in the weight loss rate curves. The first weight loss between 200 and 275 °C could be due to the decomposition of unbound CTAC molecules, as seen in Supplementary Fig. 14a; the other two losses can be ascribed to the decomposition of adsorbed CTAC molecules, reflecting the bilayer structure of the CTAC molecules on the NPs^{3,4}. Therefore, we consider the sum of the last two weight losses to be the residual amount of CTAC molecules on the washed Pd NPs. The Pd_(Br) and Pd_(Cl) NPs exhibit weight losses of CTAC of 49.3 % and 25.0 %, respectively, indicating that the amount of CTAC molecules adsorbed on the Pd_(Br) specimen is approximately twice as high as on the Pd_(Cl) specimen. This trend is consistent with the APT results, which revealed a larger amount of CTA⁺ ions on Pd_(Br) than on Pd_(Cl) NPs after washing.

Supplementary Figure 15. XPS analyses. Pd 3d core level spectra of **a** as-synthesized Pd_(Br) NPs, **b** washed Pd_(Br) NPs, **c** as-synthesized Pd_(Cl) NPs, **d** washed Pd_(Cl) NPs.

Supplementary Figure 15 shows the Pd 3d core level spectra of as-synthesized and washed Pd_(Br) and Pd_(Cl) NPs. Two major peaks (fitted by red and blue lines) are observed in the spectra. The peak at 335.5 – 335.8 eV (red) is due to metallic Pd (Pd⁰), while the peak at 337.3 – 337.5 eV (blue) can be ascribed to Pd atoms chemically bound to electronegative elements, such as O, Br, and Cl⁵. Since both NP specimens were synthesized in a surfactant-rich environment, the contribution of Pd oxide to the peak at 337.3 – 337.5 is low (see orange and green lines, corresponding to PdO and PdO₂ peaks, respectively⁶). Therefore, we can assign the peaks fitted by the blue solid lines to PdBr₂ and PdCl₂ for the Pd_(Br) and Pd_(Cl) specimens, respectively.

Both Pd_(Br) and Pd_(Cl) specimens exhibit a more intense Pd⁰ peak and a less intense Pd–halide peak after washing. Furthermore, as shown in Supplementary Table 3, the intensity ratio between the Pd–halide complex and Pd⁰ peak was always higher for Pd_(Br) than for Pd_(Cl). These results show that Pd_(Br) NPs were more densely covered with ligands than Pd_(Cl) in good agreement with the APT results.

Supplementary Table 3. Comparison of the area ratio (%) of Pd peaks for different NPs.

	As-synthesized Pd _(Br)	Washed Pd _(Br)	As-synthesized Pd _(Cl)	Washed Pd _(Cl)
Pd metal (Pd⁰)	12.7	22.1	24.5	69.2
Pd-halides	66.0	58.4	59.1	16.9
PdO	-	-	-	8.9
PdO₂	21.3	19.6	16.4	5.1
Pd-halides/Pd⁰	5.19	2.64	2.41	0.24

Supplementary References

3. Nikoobakht, B. & El-Sayed, M. A. Evidence for bilayer assembly of cationic surfactants on the surface of gold nanorods. *Langmuir* **17**, 6368–6374 (2001).
4. Sui, Z. *et al.* An improved approach for synthesis of positively charged silver nanoparticles. *Chem. Lett.* **34**, 100–101 (2005).
5. Kumar, G., Blackburn, J. R., Albridge, R. G., Moddeman, W. E. & Jones, M. M. Photoelectron Spectroscopy of Coordination Compounds. II. Palladium Complexes. *Inorg. Chem.* **11**, 296–300 (1972).
6. Kim, K. S., Gossman, A. F. & Winograd, N. X-ray photoelectron spectroscopic studies of palladium oxides and the palladium-oxygen electrode. *Anal. Chem.* **46**, 197–200 (1974).

Comment 4:

“p.7, line 137-139. Concerning long-term storage experiments, authors state that for a better comparison, KCl has been added to Pd(Br) sample, in order to realize the same concentration of chloride anions (etching agent) as in Pd(Cl). This approach is reasonable, however it should be also noted that the addition of KCl to Pd(Br) leads to an increase of salinity and ionic strength, which is not experienced in the case of Pd(Cl). According to salting-out effect, the addition of electrolyte results in a decreased solubility of gases (specifically oxygen) in water and the decrease is proportional to the salt concentration and ionic strength. The decreased amount of dissolved oxygen might contribute to the observed attenuation of oxidative etching in the case of Pd(Br), could then authors add a comment in the text about this important point, please?”

Response 4:

We thank the reviewer for this excellent comment. To examine the salting-out effect, we performed another set of long-term storage experiments. This time we added KCl to the washed Pd_(Br) sample and KBr to the washed Pd_(Cl) sample to match the salt concentrations; we made very similar observations as in the previous experiment, *i.e.*, the Pd_(Br) NPs were well preserved, whereas the Pd_(Cl) NPs were largely dissolved (see Supplementary Figure 3). Thus, we can attribute the difference in the oxidation resistance of the samples to the difference in the ligand distribution on the Pd surface. We added the sentence below to the manuscript and added Supplementary Figure 3 to the supporting information.

Supplementary Figure 3. TEM images of washed Pd NPs with matched halide ion concentrations left for ten days in air. a Pd_(Br) and b Pd_(Cl) NPs. (Scale bars: 200 nm in a, c and 20 nm in b, d)

“Furthermore, to consider the difference in oxygen solubility due to different salt concentrations in the Pd_(Br) and Pd_(Cl) specimens, we performed another set of control experiments. We added identical molar concentrations of KCl to the washed Pd_(Br) and KBr to the washed Pd_(Cl) NPs to match the salt concentrations and examined the specimens with TEM after ten days of exposure to air. We made very similar observations as in the previous experiments, namely the Pd_(Br) NPs were well preserved, whereas the Pd_(Cl) NPs were largely dissolved (see Supplementary Fig. 3). Thus, the difference in oxidation resistance of the Pd_(Br) and Pd_(Cl) NPs could be indeed ascribed to the difference in concentrations of ligands on the Pd surface.” (page 8, line 146)

Comment 5:

“PDOS calculations demonstrated that Pd forms stronger interactions with chloride anions than with bromide ones, as stated at p.14 lines 265-270. However, at p.19 lines 361-362 authors state that stronger bonds are formed between Pd and Br-. Can the author clarify this point, please? Is it correct to classify the interaction between Pd surface atoms and halide anions as covalent bonds?”

Response 5:

Thank you for this comment.

Pd forms stronger interactions with chloride anions than bromide anions in vacuum. This result was obtained from DFT calculations, which considered supercells in vacuum and at zero Kelvin. However, in an aqueous solution the reverse behavior is observed due to the influence of the solvation energy and electron affinity. As stated on page 16, line 302 – 303, given that the actual synthesis of Pd_(Br) and Pd_(Cl) NPs was performed in aqueous solutions, the Born-Haber cycle approach (see Supplementary Figure 18) should be considered.

According to the Born-Haber cycle approach, the binding energy (E_B) of a halide ion (X^-) to a Pd surface is given as $E_B = BE_x - \Delta H_{sol} - \Phi_{Pd} + EA_x$, where BE_x is the binding energy for the adsorption of halogen atoms to a negatively charged Pd surface with one extra electron. ΔH_{sol} , Φ_{Pd} , and EA_x are the solvation energy of X^- in water, the work function of the corresponding Pd surface, and the electron affinity of X, respectively. Upon applying the Born-Haber cycle, the adsorption strengths of Cl^- and Br^- ions exhibit an opposite trend in terms of BE_x and E_B values as compared to the vacuum state (see Fig. 4b).

We use the term ‘covalent’ to describe the interaction between Pd surface atoms and halide anions, based on the PDOS band overlap between Pd surface and halide anions. We are also well aware that the chemical bonding between a Pd surface and halide anion cannot be viewed as 100% covalent. To clarify this point, we modified the sentence as shown below.

Original manuscript

“We revealed that Br^- ions form stronger covalent bonds with Pd than Cl^- ions and enhance the adsorption of CTA⁺ ions on Pd NP through electrostatic interactions.”

Revised manuscript

“We revealed that under the given synthesis conditions, Br⁻ ions are more strongly chemisorbed on the Pd surface than Cl⁻ ions and enhance the adsorption of CTA⁺ ions on Pd NP through electrostatic interactions.” (page 21, line 389)

Comment 6-1:

“p.20, Synthesis of Pd nanoparticles. It is not declared which is the reducing agent for metal nanoparticle formation. Could the authors add this important information, please?”

Response 6-1:

We are thankful to the reviewer for this comment.

According to (Lee et al. J. Am. Chem. Soc. 131.47 (2009): 17036-17037; Kang et al. ACS Nano 7.9 (2013): 7945-7955), CTAC can serve both as a mild reducing agent and a stabilizer. Therefore, CTAC most likely acted as a reducing agent in our synthesis method. In order to clarify which of the used chemicals acted as a reducing agent, we first tried to synthesize Pd_(Br) and Pd_(Cl) nanoparticles without CTAC. Next, we synthesized the nanoparticles without KBr/KCl under otherwise identical conditions. No particles were formed in the absence of CTAC, indicating that CTAC indeed served as a reducing agent.

We modified the following sentence and added the reference below to the manuscript.

“K₂PdCl₄ and KBr were used as a precursor and a shape-controlling agent, respectively. CTAC was added to serve both as a surfactant and a mild reducing agent⁶⁷.” (page 22, line 409)

References

67. Kang, S. W. *et al.* One-pot synthesis of trimetallic Au@PdPt core-shell nanoparticles with high catalytic performance. *ACS Nano* 7, 7945–7955 (2013).

Comment 6-2:

“Why did authors use CTAC also for the synthesis of Pd(Br)? CTAB (the cetrimonium bromide salt) is largely available and likely its use would have been preferable in the case of Pd(Br) NPs.”

Response 6-2:

We agree with the reviewer. In principle, the Pd_(Br) NPs could also be synthesized with CTAB. However, the synthesis of the Pd_(Br) NPs would involve Cl⁻ ions anyway because the precursor, K₂PdCl₄, contains Cl⁻ ions. In order to avoid some unexpected/unwanted variables, such as different impurity contents of the used CTAC and CTAB, and for a more reliable direct comparison of the samples, we decided to use CTAC for both Pd_(Br) and Pd_(Cl).

Comment 7:

“Concerning the title, a more detailed version is suggested (e.g. “Three-dimensional Atomic Mapping of Ligands on Palladium Nanoparticles by atom probe tomography”)”

Response 7:

We thank the reviewer for this suggestion. We have modified the title of the paper to:

“Three-dimensional Atomic Mapping of Ligands on Palladium Nanoparticles by Atom Probe Tomography”

Comment 8:

“The introduction is well written, it contains relevant references to the literature and clearly explains which are the main challenges in the field and the novelty of the work. I just suggest introducing more recent references on the role of halides in regulating the chemistry of metal nanoparticles (e.g. Nano Research, 2017, 10(3), 1064–1077; Nanoscale, 2019, 11, 15612–15621; Nanoscale, 2017, 9, 17914–17921; Nanoscale, 2016, 8, 3962–3972; PHYSICAL REVIEW MATERIALS, 2020, 4, 096001; ChemNanoMat 2020, 6, 576-588.)”

Response 8:

We thank the reviewer for this suggestion. We have added all the references suggested by the reviewer to the introduction part:

“Although each of these methods is unique, they share a common feature, i.e., they all rely on capping ligands. Capping ligands are additives adsorbed on specific crystallographic surfaces of the NPs; they can prevent NP agglomeration and control the NP size, shape, and functionality^{13–16}. Therefore, capping ligands are paramount to tune the properties of colloidal NPs^{17–18}. The most used capping ligands are thiols, block copolymers, cetrimonium, and halide ions¹⁹; the latter two are particularly advantageous as they can be applied in various nanoparticle systems^{20–22}.”

14. Zhang, J. et al. PdPt bimetallic nanoparticles enabled by shape control with halide ions and their enhanced catalytic activities. *Nanoscale* 8, 3962–3972 (2016).

15. Qiu, P., Lian, S., Yang, G. & Yang, S. Halide ion-induced formation of single crystalline mesoporous PtPd bimetallic nanoparticles with hollow interiors for electrochemical methanol and ethanol oxidation reaction. *Nano Res.* 10, 1064–1077 (2017).

18. Löfgren, J., Rahm, J. M., Brorsson, J. & Erhart, P. Computational assessment of the efficacy of halides as shape-directing agents in nanoparticle growth. *Phys. Rev. Mater.* 4, 1–9 (2020).

20. King, M. E. & Personick, M. L. Defects by design: Synthesis of palladium nanoparticles with extended twin defects and corrugated surfaces. *Nanoscale* 9, 17914–17921 (2017).

21. King, M. E., Kent, I. A. & Personick, M. L. Halide-assisted metal ion reduction: Emergent effects of dilute chloride, bromide, and iodide in nanoparticle synthesis. *Nanoscale* 11, 15612–15621 (2019).

22. Yang, T. H., Zhou, S., Zhao, M. & Xia, Y. Quantitative Analysis of the Multiple Roles Played by Halide Ions in Controlling the Growth Patterns of Palladium Nanocrystals. *ChemNanoMat* 6, 576–588 (2020).

REVIEWERS' COMMENTS

Reviewer #1 (Remarks to the Author):

Authors did an outstanding job at addressing all of my concerns and this paper is now ready to be accepted.

Reviewer #2 (Remarks to the Author):

The reviewer appreciated the revision work made by the Authors. This paper represents a significant new contribution and it should be now published as is on Nature Communications.